🔓 | **Open Peer Review** | Computational Biology | Research Article

# Modeling dynamic oxygen permeability as a mechanism to mitigate oxygen-induced stresses on photosynthesis and N$_2$ fixation in marine *Trichodesmium*

**Weicheng Luo,**[1,2,3] **Keisuke Inomura,**[4] **Ondřej Prášil,**[2] **Meri Eichner,**[2] **Ya-Wei Luo**[1]

**ABSTRACT** *Trichodesmium*, the predominant marine diazotrophic cyanobacterium, concurrently performs nitrogen (N$_2$) fixation and photosynthesis, the latter of which produces oxygen (O$_2$) that inhibits N$_2$ fixation. Hopanoid lipids in *Trichodesmium* may play a role in dynamically regulating membrane permeability to O$_2$, potentially alleviating O$_2$ stress on N$_2$ fixation. However, the physiological impacts of this dynamic permeability are not well understood. We developed a model showing that dynamically modulating membrane O$_2$ permeability can enhance N$_2$ fixation and growth of *Trichodesmium* by over 50%. High O$_2$ permeability ($1.5 \times 10^{-4}$ of O$_2$ diffusivity in seawater) during strong photosynthesis accelerates O$_2$ exhaust, reducing energy-consuming photorespiration by ~40%, while low O$_2$ permeability ($1.0 \times 10^{-5}$ diffusivity) during active N$_2$ fixation minimizes O$_2$ stress on N$_2$ fixation. Together, these mechanisms increase the carbon and iron use efficiencies by ~70%. Our study provides a mechanistic and quantitative framework for how dynamic O$_2$ permeability benefits *Trichodesmium*, offering insights potentially applicable to other diazotrophs.

**IMPORTANCE** *Trichodesmium* is a key player in marine N$_2$ fixation, essential for oceanic productivity and global biogeochemical cycles. However, a significant challenge arises from the concurrent photosynthetic production of O$_2$ during N$_2$ fixation, which can inhibit N$_2$ fixation and cause energy-wasting photorespiration. We develop a physiological model showing that *Trichodesmium* may dynamically regulate membrane O$_2$ permeability to enhance N$_2$ fixation and growth. The model suggests two mechanisms: elevated O$_2$ permeability during the early daytime of strong photosynthesis accelerates O$_2$ exhaust to the environment, reducing photorespiration, while reduced O$_2$ permeability later limits O$_2$ influx from the environment, lowering wasteful respiration and maintaining a low intracellular O$_2$ level for active N$_2$ fixation. These adaptations improve the efficiency of carbon and iron utilization, thereby facilitating N$_2$ fixation and growth in *Trichodesmium*. This study sheds light on how *Trichodesmium* and other N$_2$-fixing microorganisms can optimize their physiological processes in response to environmental challenges.

**KEYWORDS** *Trichodesmium*, dynamic oxygen permeability, nitrogen fixation, photorespiration, respiratory protection

*T*richodesmium is a major photoautotrophic contributor to marine nitrogen (N$_2$) fixation (1–3). *Trichodesmium* faces physiological challenges, such as the decrease in the activity of nitrogenase (the enzyme for N$_2$ fixation) upon exposure to oxygen (O$_2$) (4–6). Given that *Trichodesmium* simultaneously conducts N$_2$ fixation and O$_2$-producing photosynthesis during the daytime (4, 7), *Trichodesmium* has developed several physiological strategies to cope with this O$_2$ stress on nitrogenase and protect N$_2$

**Peer Reviewer** Nir Keren, Hebrew University of Jerusalem, Jerusalem, Israel

Address correspondence to Weicheng Luo, weicheng123@stu.xmu.edu.cn, or Ya-Wei Luo, ywluo@xmu.edu.cn.

The authors declare no conflict of interest.

fixation (8). One of these is the respiratory protection: *Trichodesmium* creates a low-$O_2$ intracellular environment to realize sufficient $N_2$ fixation by temporally segregating photosynthesis and $N_2$ fixation and wastefully respiring organic carbon with intracellular $O_2$ (9–11). A potentially complementary strategy is the diffusion adjustment; a proper low cell membrane permeability to $O_2$ can contribute to forming and maintaining the low-$O_2$ window (10, 12, 13).

Notably, during the early light period with high rates of photosynthesis and $O_2$ production, low cell permeability to $O_2$ could lead to high intracellular $O_2$ concentrations, resulting in oxidative stress on photosynthesis as well as photorespiration, an energy-inefficient consumption of $O_2$ (10, 13, 14). Photorespiration is a light-dependent process that consumes ATP (adenosine triphosphate), NADPH (nicotinamide adenine dinucleotide phosphate hydrogen), reduced N and $O_2$, and produces $CO_2$ (15). This oxygenation reaction is catalyzed by RuBisCO (ribulose-1,5-bisphosphate carboxylase/oxygenase) with RuBP (ribulose-1,5-bisphosphate) and $O_2$ as substrates (15). Consequently, the high intracellular $O_2$ concentration during early daytime can compete with $CO_2$ and inhibit the carboxylation activity of RuBisCO, thus increasing photorespiration and reducing photosynthetic carbon fixation (13, 14, 16). While cyanobacteria are known to operate carbon concentrating mechanisms (CCM) to increase intracellular $CO_2$ concentration and thus shift the $CO_2{:}O_2$ ratio in favor of carboxylation rather than oxygenation, recent studies have suggested that photorespiration may still play an important role in these organisms (17). Technical difficulties have so far hindered direct quantification of photorespiration rates in cyanobacteria, and given these uncertainties, photorespiration has not been explicitly resolved in previous models of *Trichodesmium* (10, 12, 13).

Due to the importance of $O_2$ concentration on the likelihood of photorespiration, the permeability of the cell membrane to $O_2$ may impact the occurrence of photorespiration. For example, a high cell permeability to $O_2$ could facilitate the rapid diffusion of intracellular $O_2$ to the extracellular environment, likely decreasing the photorespiration rate during the early light period when photosynthesis is strong in *Trichodesmium* (14). This high $O_2$ permeability could also enhance the diffusion rate of extracellular $O_2$ into the cytoplasm during the low-$O_2$ window, thereby elevating the respiratory protection required to consume organic carbon and intracellular $O_2$ as an indirect cost for $N_2$ fixation (10). On the contrary, a low cell permeability to $O_2$ would elevate the photorespiration but benefit $N_2$ fixation. The above scenarios are based on the assumption of diurnally constant cell permeability to $O_2$, as employed in previous model studies (10, 12, 13). However, *Trichodesmium* can synthesize hopanoids, which can be intercalated into lipid bilayers of membranes (18, 19). The planar and hydrophobic structure of hopanoids may decrease the membrane permeability to $O_2$. Also, hopanoids may form rafts (high concentration domain), which can be distributed within the membrane, and thus may dynamically regulate cell permeability to $O_2$ (dynamic-permeability model case) (18, 20). Such a dynamic regulation is currently a hypothesis; it is likely that the dynamic expression of hopanoid biosynthesis genes may occur, given the highly dynamic protein expression in *Trichodesmium* (21). The potential physiological implications of this dynamic cell permeability to $O_2$ ($DPO_2$) to *Trichodesmium* remain poorly understood and warrant further investigation.

In this study, we hypothesize that $DPO_2$ helps to regulate intracellular $O_2$ levels and mitigates $O_2$-induced stresses in *Trichodesmium*, promoting the efficiency of key enzymes such as RuBisCO and nitrogenase in the context of temporally segregating the activities of the two enzymes. To test the hypothesis, we improved previous models by representing more processes, including photorespiration and $DPO_2$. The analyses of the model results, along with the comparison to additional experiments of fixed $O_2$ permeability, which was set diurnally constant (fixed-permeability model case), provided a mechanistic and quantitative understanding of the potential role of $DPO_2$ in impacting photorespiration and $N_2$ fixation in *Trichodesmium*.

## MATERIALS AND METHODS

The model in this study was developed by incorporating new representations of photorespiration and $DPO_2$ into previous models (10, 22). In the subsequent sections, we present a concise overview of the model structure. We provide more detailed descriptions, parameter values, intermediate variables, and state variables in Supplementary Methods and Table S1 to S3.

### General model framework

The model (Fig. 1A) simulates key physiological processes in *Trichodesmium* trichome, including photosynthetic electron transfer (PET), carbon fixation, photorespiration, and $N_2$ fixation over a 12 hour diurnal cycle. These processes are modulated by the dynamic allocation of Fe, ATP, and NADPH to different metabolic pathways, as well as by intracellular $O_2$ management (Fig. 1A). Two pathways of PET are represented, including linear PET (LPET) that produces ATP and NADPH, and alternative electron transfer (AET) that only generates ATP. ATP and NADPH are used in various processes as described below. $N_2$ fixation occurs only when intracellular $O_2$ is low. Photorespiration increases with increasing intracellular $O_2$ levels.

### Photosynthetic pathways

The photosynthetic pathways followed previous model schemes (10, 22). The total rate of PET is positively regulated by light intensity and the Fe allocated to photosystems. Conversely, it is mitigated by respiratory protection mechanisms (4). The proportion of electrons directed toward linear PET (LPET) and alternative electron transport (AET) is computed at each time step. This is assumed to meet the immediate intracellular requirements for ATP and/or NADPH (10, 22).

### $O_2$ production and dynamic permeability

$O_2$ is exclusively produced by LPET (23, 24). AET reduces photosynthetic $O_2$ production while also supporting ATP production (23–25). In addition, *Trichodesmium* can perform respiratory protection to wastefully consume organic carbon and intracellular $O_2$ and protect $N_2$ fixation (4, 10, 12, 26, 27). $O_2$ can also physically diffuse between cytoplasm and extracellular environment (Fig. 1A). The direction and rate of $O_2$ diffusion depend on the difference between intracellular and extracellular $O_2$ concentration, as well as the $O_2$ permeability of the cell membrane ($\varepsilon$) which is represented as the relative diffusivity to seawater (28).

In the "dynamic-permeability" model case, the membrane $O_2$ permeability is assumed to vary diurnally (18). The membrane $O_2$ permeability is parameterized to increase with intracellular $O_2$ concentration using a Michaelis-Menten equation (Fig. 1B):

$$\varepsilon = \varepsilon_{\max} \cdot \frac{O_2}{O_2 + k_{O_2}^{\mathrm{diff}}}, \tag{1}$$

where $\varepsilon_{\max}$ is the maximal relative diffusion coefficient of $2.0 \times 10^{-4}$, and $k_{O_2}^{\mathrm{diff}}$ (0.213 mol $O_2$ m$^{-3}$), a saturating concentration in seawater at 34 PSU salinity and 25°C (29), is the half-saturation constant of $O_2$ for relative diffusion coefficient.

In addition, to quantitatively assess the physiological roles of $DPO_2$, we performed another "fixed-permeability" model case with a fixed $\varepsilon$ of $1.0 \times 10^{-4}$.

Given $\varepsilon$, the rate of $O_2$ diffusion $\left( T_{O_2,\ \mathrm{mol\ O_2\ m^{-3}s^{-1}}} \right)$ between the intracellular cytoplasm and the extracellular environment is calculated using the scheme proposed by Staal et al. (28) for cylinder-shaped cells:

$$T_{O_2} = \frac{-2 \cdot \Pi \cdot d_{O_2} \cdot L}{V} \cdot \left\{ \frac{1}{\varepsilon} \cdot \ln\left( \frac{R}{R + L_g} \right) - \ln\left( \frac{R + L_g + L_b}{R + L_g} \right) \right\}^{-1} \cdot \left( O_2^E - O_2 \right), \tag{2}$$

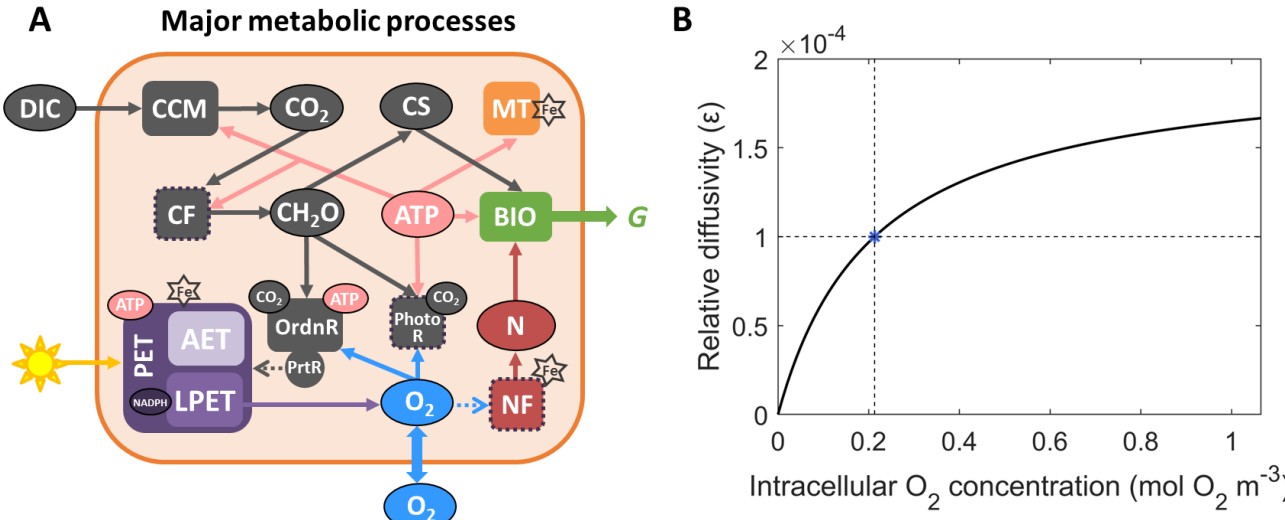

**FIG 1** Schematic of the physiological model with potential dynamic $O_2$ permeability in *Trichodesmium*. (A) Photosynthesis, photorespiration (PhotoR), $N_2$ fixation, and other key processes are simulated within the *Trichodesmium* trichome. The pentagrams marked Fe-requiring processes, with fundamental intracellular Fe pools shown in Fig. S1A. Dashed arrows represent inhibition effects. CF: carbon fixation; NF: $N_2$ fixation; PrtR: respiratory protection; OrdnR: ordinary respiration; $CH_2O$: carbohydrate; CS: carbon skeleton; N: fixed nitrogen; MT: maintenance; BIO: biosynthesis; G: growth rate. (B) The model parameterizes the dynamic $O_2$ permeability by assuming that the intracellular $O_2$ concentration regulates the $O_2$ diffusion coefficient of the cell membrane relative to that of seawater. Blue asterisk denotes the relative diffusivity of the cell membrane at the half-saturating coefficient of intracellular $O_2$ concentration (0.213 mol $O_2$ m$^{-3}$). Note that in the fixed-permeability model case, the reference value of $\varepsilon = 1.0 \times 10^{-4}$ was selected based on observational constraints from prior studies (10, 22).

where $d_{O_2}$ is the $O_2$ diffusion coefficient in seawater at 34 PSU and 25°C, $L$ (m) and $V$ (m$^3$) represent the length and the volume of the trichome, $R$ (m) represents the radius of the cytoplasm, $L_g$ (m) denotes the thickness of the cell membrane, $L_b$ (m) denotes the thickness of the boundary layer, and $O_2^E$ refers to the extracellular far-field $O_2$ concentration.

## Photorespiration

Our model additionally represents photorespiration. Photorespiration requires energy using organic carbon and $O_2$ as substrates (14, 30). The energy usage by photorespiration consequently reduces the energy availability for carbon and $N_2$ fixation. The maximal photorespiration rate $\left[V_{PR}^{max}, \text{ mol C (mol C)}^{-1}\text{s}^{-1}\right]$ is computed based on the assumption that ATP produced by PET is fully consumed by photorespiration:

$$V_{PR}^{max} = \frac{V_{ATP}}{q_{PR}^{ATP}}, \qquad (3)$$

where $q_{PR}^{ATP} = 7$ mol ATP (mol C)$^{-1}$ is the ATP to C ratio in photorespiration (15).

The rate of photorespiration $\left[V_{PR}, \text{ mol C (mol C)}^{-1}\text{s}^{-1}\right]$ is also regulated by substrates, including intracellular carbohydrate $[CH_2O, \text{ mol C (mol C)}^{-1}]$ and $O_2$ $[O_2, \text{ mol } O_2 \text{ m}^{-3}]$:

$$V_{PR} = V_{PR}^{max} \cdot \frac{CH_2O}{CH_2O + K_{CH_2O}^{PR}} \cdot \frac{O_2}{O_2 + K_{O_2}^{PR}} \qquad (4)$$

where $k_{CH_2O}^{PR} = 0.4$ [mol C (mol C)$^{-1}$] and $k_{O_2}^{PR} = 1.92$ (mol $O_2$ m$^{-3}$) are half-saturating coefficients of $CH_2O$ and $O_2$ for photorespiration (Fig. S1B).

The NADPH, ATP, and $O_2$ consumption rates of photorespiration [$V_{NADPH}^{PR}$, $V_{ATP}^{PR}$, $V_{O_2}^{PR}$, mol NADPH (mol C)$^{-1}$ s$^{-1}$, mol ATP (mol C)$^{-1}$ s$^{-1}$, and mol O2 (mol C)$^{-1}$ s$^{-1}$ are:

$$V_{NADPH}^{PR} = V_{PR} \cdot q_{PR}^{NADPH}, \tag{5}$$
$$V_{ATP}^{PR} = V_{PR} \cdot q_{PR}^{ATP}, \tag{6}$$
$$V_{O_2}^{PR} = V_{PR} \cdot q_{C, PR}^{O_2}, \tag{7}$$

where $q_{PR}^{NADPH}$ = 4 mol NADPH (mol C)$^{-1}$ and $q_{C, PR}^{O_2}$ = 3 mol $O_2$ (mol C)$^{-1}$ are NADPH to C and $O_2$ to C ratios in photorespiration (15).

## N$_2$ fixation

N$_2$ fixation was calculated according to previous model schemes (10, 22). N$_2$ fixation necessitates the utilization of both ATP and NADPH (31, 32). The maximal potential of N$_2$ fixation rate $\left[V_{NF}^{max}, \text{ mol N (mol C)}^{-1} \text{ s}^{-1}\right]$ occurs when the produced ATP and NADPH from PET are completely consumed by N$_2$ fixation (10). The rate of N$_2$ fixation $\left[V_{NF}, \text{ mol N (mol C)}^{-1} \text{ s}^{-1}\right]$ (see Supplementary Methods) is limited by the Fe allocated to nitrogenase $\left[\text{Fe}_{NF}, \mu\text{mol Fe (mol C)}^{-1}\right]$ (33–35) and can be impended by intracellular $O_2$, with the rate decreasing upon exposure to $O_2$ (36).

$$V_{NF} = V_{NF}^{max} \cdot \frac{\text{Fe}_{NF}}{\text{Fe}_{NF} + k_{Fe}^{NF}} \cdot \left(1 - \frac{O_2}{O_2 + k_{O_2}^{NF}}\right), \tag{8}$$

where $k_{Fe}^{NF}$ [$\mu$mol Fe (mol C)$^{-1}$] and $k_{O_2}^{NF}$ (mol $O_2$ m$^{-3}$) are half-saturating coefficients of Fe$_{NF}$ and $O_2$ for N$_2$ fixation.

Respiratory protection also followed previous model schemes (10, 22). Respiratory protection is a mechanism that involves the wasteful respiration of carbohydrates to reduce intracellular $O_2$ concentration, thereby supporting N$_2$ fixation (12). The rate of respiratory protection increases in response to the demand for N$_2$ fixation, while it decreases as the intracellular $O_2$ level rises (10, 22).

## Carbon fixation

Carbon fixation was computed upon previous models (10, 22) with a minor change by considering photorespiration. Similarly to N$_2$ fixation, carbon fixation also relies on the availability of both NADPH and ATP (37). To calculate the carbon fixation rate, the total production of NADPH and ATP at each time step is assumed to be promptly and completely utilized by intracellular processes (10, 22), including photorespiration, CCM, carbon fixation, N$_2$ fixation, and maintenance (10, 22).

Carbohydrates, which are generated through carbon fixation, stimulate the production of carbon skeletons. However, this production is subsequently downregulated due to the accumulation of these carbon skeletons (see Supplemental methods).

## Intracellular Fe pools and translocation

This part was conducted based on previous model schemes (10, 22). The total intracellular Fe, which encompasses both metabolism and storage (Fig. S1A), is calculated using a previously established scheme (33). Metabolic Fe includes Fe in photosystems, nitrogenase, maintenance, and buffer (Fig. S1A). Fe utilized by the photosystems and nitrogenase is from the buffer pool (Fig. S1A). The parameterization of synthesis and decomposition rate of photosystems, as well as the synthesis rate of active nitrogenase and its inactivation, is based on a recent model study featuring diurnally dynamic Fe allocation (22).

Considering that $DPO_2$ may contribute to improving the efficiency of Fe in photosystems and nitrogenase, thus regulating intracellular Fe allocation, model cases were run under various Fe levels for comparison.

## Model parameter values

In both fixed and dynamic $O_2$ permeability model cases, four parameters were optimized to maximize the growth rate of *Trichodesmium* (38). These parameters include the maximal respiratory protection rate $(v_{\mathrm{RP}}^{\max})$, the maximal synthesis $\left(T_{\mathrm{PS}_{\max}}^{\mathrm{BF}}\right)$ and decomposition $\left(T_{\mathrm{BF}_{\max}}^{\mathrm{PS}}\right)$ rates of photosystems, and the maximal synthesis rate $\left(T_{\mathrm{NF}_{\max}}^{\mathrm{BF}}\right)$ of nitrogenase (Table S1). The optimization was performed employing the global optimizer MultiStart in MATLAB.

Other parameters (Table S2) were either adopted from previous studies or tuned to fit the observed growth rates, $N_2$ fixation rates, and diurnal Fe in photosystems and nitrogenase from a laboratory culture experiment (34). Given that the experiment was conducted under constant light intensity (90 µmol m$^{-2}$ s$^{-1}$), we adopted the same light intensity as $DPO_2$. After tuning model parameters (Table S2), the modeled growth rates (0.28 and 0.45 d$^{-1}$ under low and high Fe, respectively) were well aligned with the observations (Table S4). Moreover, the model reproduced diurnal patterns of photosystem and nitrogenase (under low and high Fe, for photosystem Fe, $R^2 = 0.13$ and 0.90, respectively; for nitrogenase Fe, $R^2 = 0.60$ and 0.80, respectively) (Fig. S2). The low $R^2$ value for photosystem Fe under the low Fe condition reflects substantial natural variability in *Trichodesmium* physiology, generally compounded by nonlinear environmental interactions (34, 35). While limited observational points (5 samples during the light period) were used for constraints, the model still captured key diurnal dynamics (Fig. S2), with $R^2$ values expected to improve through higher-resolution sampling. In addition, the reliability index (RI) (Equation 9) (39, 40) was calculated to further evaluate the performance of the model compared to observations (under low and high Fe, for photosystem Fe, RI = 1.04 and 1.02, respectively; for nitrogenase Fe, RI = 1.14 and 1.01, respectively). These RI levels close to 1.0 indicate strong consistency between observations and model results, supporting the robustness of our model (39, 40).

$$\mathrm{RI} = \exp\left(\sqrt{\frac{1}{n}\sum_{i=1}^{n}\left(\ln\frac{\mathrm{Observation}_i}{\mathrm{Model}_i}\right)^2}\right), \tag{9}$$

where Observation and Model are observations and model results, respectively; *n* is the number of observational data points.

Note that the constant light intensity was only used when tuning model parameters. All the model results presented in the following were simulated with dynamic light intensity using a sine function over a 12 hour light period (41).

## RESULTS

### Simulated growth rate, carbon and $N_2$ fixation rates, and $O_2$ concentration

Similar to a previous model study (22), the simulations encompassed 10 Fe′ levels (20–1,800 pM) to approximately represent the *Trichodesmium* Fe quota of 10–1,000 µmol Fe (mol C)$^{-1}$, a range observed in field *Trichodesmium* samples (33). Our model showed a positive correlation between Fe concentration and both $N_2$ fixation and growth rates (Fig. S3), consistent with trends from previous culturing experiments of *Trichodesmium* (35). In addition, model results exhibited that the influence of $DPO_2$ on promoting growth rates decreased from 78% to 33% as Fe concentration increased (Fig. S3), indicating that the physiological benefits of $DPO_2$ in *Trichodesmium* were more pronounced under low Fe conditions. In the following, we focused on analyzing the model results at two levels of dissolved inorganic Fe (40 pM and 1,250 pM) that were set

in the laboratory experiments (34). Under these two Fe levels, $DPO_2$ promoted modeled growth rates of *Trichodesmium* by 61% and 30%, respectively.

Our results revealed that while the dynamic-permeability case exhibited higher growth rates than the fixed-permeability case under both Fe conditions, the gross carbon fixation rates in the dynamic-permeability case were even lower (Fig. 2). This implies that dynamic $O_2$ permeability led to an improvement in carbon use efficiency of *Trichodesmium*, defined here as the ratio of net to gross carbon production (Fig. 2). The decrease in the requirement for carbon fixation and the increase in carbon use efficiency is attributed to the lowered photorespiration and the downregulated requirement of respiratory protection (see Discussion) (Fig. 2). Furthermore, our simulation results aligned with previous studies (22, 42, 43), demonstrating higher carbon use efficiency under higher Fe conditions.

$DPO_2$ benefits modeled *Trichodesmium* via modulating carbon and $N_2$ fixation rates and intracellular $O_2$ levels, with the physiological roles of $DPO_2$ differing in two periods (Fig. 3). We first analyzed results under low-Fe conditions.

During the early period, compared to the fixed-permeability case, the carbon fixation rate in the dynamic-Fe case was slightly lower with a decreasing pattern (approximately 1–4 h) (Fig. 3A). This indicates that $DPO_2$ could lower the requirement for carbon storage. In addition, lower intracellular $O_2$ level in the dynamic-permeability case (Fig. 3E and G) reduced the carbon consumption by photorespiration (see Discussion).

During the low-$O_2$ window, the $N_2$ fixation rate in the dynamic-permeability case was higher (Fig. 3C). A wider low-$O_2$ window was presented in the dynamic-permeability case (Fig. 3E), suggesting $DPO_2$ could reduce the stress from $O_2$ on $N_2$ fixation (see Discussion).

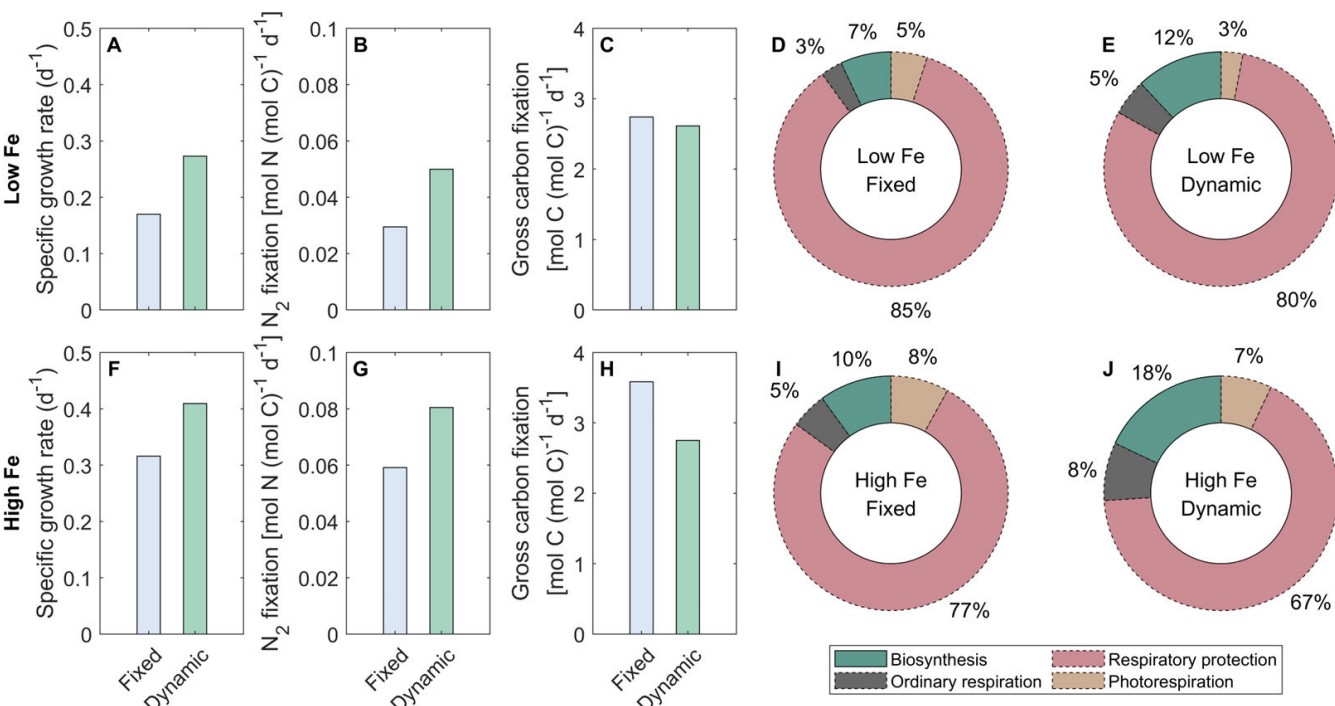

**FIG 2** Modeled daily-integrated results of *Trichodesmium*. The model is simulated with diurnally fixed and dynamic $O_2$ permeability of the cell membrane under low-Fe (40 pM) and high-Fe (1,250 pM) conditions. Model results include growth rates (A, F), $N_2$ fixation rates (B, G), and gross carbon fixation rates (C, H). The number in the inner circle represents the daily-integrated gross carbon fixation rate (mol C $[$mol C$]^{-1}$ $d^{-1}$). The fixed carbon is allocated to photorespiration, respiratory protection, ordinary respiration, and biosynthesis. Carbon use efficiency: the fraction of gross fixed carbon allocated to biosynthesis, which is highlighted using solid lines (D, E, I, and J).

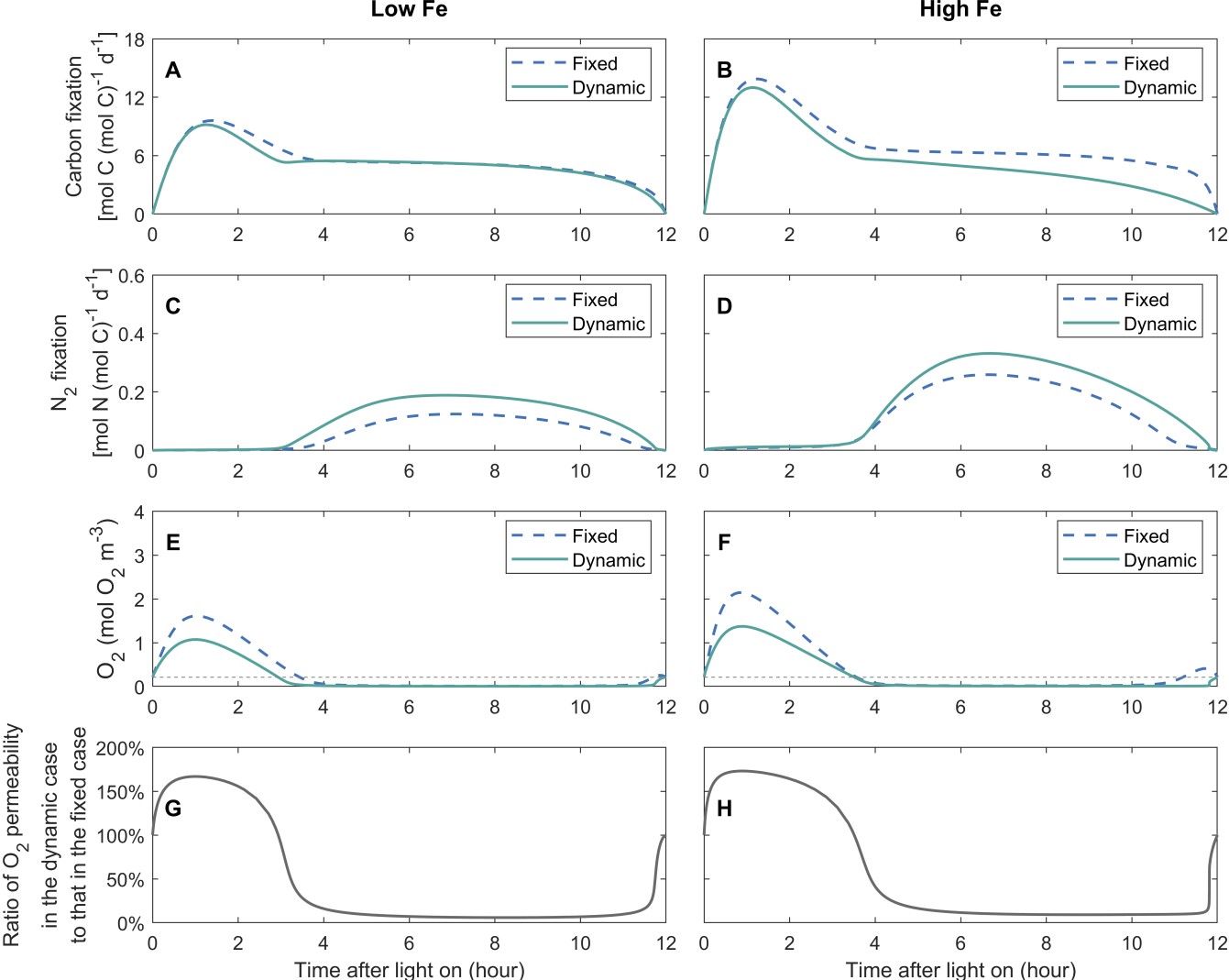

**FIG 3** Simulated instantaneous rates of gross carbon fixation and $N_2$ fixation, intracellular $O_2$ concentrations, and $O_2$ permeability of the cell membrane during the light period. The $O_2$ permeability is shown as the ratio of its values in the dynamic-permeability case to those in the fixed-permeability case. The model is simulated with diurnally fixed and dynamic $O_2$ permeability of the cell membrane under low-Fe (40 pM) (A, C, E, and G) and high-Fe (1250 pM) (B, D, F, and H) conditions. The thin black dashed lines represent the ambient far-field $O_2$ concentration.

Under the high-Fe condition, diurnal patterns of modeled carbon and $N_2$ fixation rates and intracellular concentrations were similar to those under the low-Fe condition, but at slightly higher levels (Fig. 3).

### Simulated diurnal intracellular $O_2$ fluxes

The physiological functions of $DPO_2$ in regulating intracellular $O_2$ fluxes varied diurnally. During the early daytime (approximately 0–3 h of the light period), the daily-integrated net $O_2$ production rate by PET was slightly lower compared to the dynamic-permeability case under low Fe (Fig. 4A), with a more pronounced difference under high Fe (Fig. 4B) (see Discussion). Intracellular $O_2$ diffused out of the cell cytoplasm (Fig. 4C and D) due to the high intracellular $O_2$ concentration (Fig. 3E and F), induced by the high net $O_2$ production rate of PET (Fig. 4A and B). The intracellular $O_2$ concentration in the dynamic-permeability case was lower than that in the fixed-permeability case (Fig. 3E and F), while the physical diffusion rates of $O_2$ in both cases were similar (Fig. 3C and D). This can be attributed to the higher $O_2$ permeability of the cell membrane in

the dynamic-permeability case (Fig. 3G and H), which facilitated quick intracellular $O_2$ diffusion into the extracellular environment.

In addition, high intracellular $O_2$ concentrations during this period stimulated photorespiration (Fig. 4E and F). The lower photorespiration rates in the dynamic-permeability case (Fig. 4E and F) were attributed to reduced intracellular $O_2$ concentration (Fig. 3E and F). Therefore, $DPO_2$ has the potential to alleviate the stress caused by photorespiration, particularly during the early daytime when the net $O_2$ production rate is high.

In the following light period, the intracellular low-$O_2$ window was created (Fig. 3E and F) due to the downregulation of the net $O_2$ production by PET (Fig. 4A and B) and the high $O_2$ consumption by respiratory protection (Fig. 4G and H) (10), allowing the occurrence of $N_2$ fixation. Extracellular $O_2$ diffused into the cytoplasm (Fig. 4C and D), which was slowed down by the lowered $O_2$ permeability in the dynamic-permeability case (Fig. 3G and H). Therefore, $DPO_2$ saved organic carbon required by respiratory protection (Fig. 4G and H) to maintain low $O_2$ levels in *Trichodesmium* (Fig. 3). In both the dynamic- and fixed-permeability cases, the photorespiration rate approached zero during this period (Fig. 4E and F), indicating that the creation of the low-$O_2$ window also helped to mitigate $O_2$-induced stresses on photosynthesis.

## DISCUSSION

In this study, we developed a physiological model of *Trichodesmium* trichome to quantitatively investigate the impact of photorespiration and the physiological advantages of $DPO_2$ (Fig. 1). Intracellular $O_2$ management such as respiratory protection was considered, with the temporal segregation between photosynthesis and $N_2$ fixation formed and the low-$O_2$ window created (Fig. 3). These were in line with previous model studies and observations (4, 9, 10, 12, 22). The model results demonstrate that $DPO_2$ enhanced *Trichodesmium* $N_2$ fixation and growth rates (Fig. 2, 3, and 5 and S3) by decreasing photorespiration and respiratory protection and increasing carbon and Fe use efficiency.

### Dynamic $O_2$ permeability lowers photorespiration

In the model case when the $O_2$ permeability was fixed, 5% and 8% of the daily-integrated gross fixed carbon were consumed by photorespiration under low-Fe and high-Fe conditions, respectively (Fig. 2, 4E, and F), which is comparable to the fractions allocated to biomass synthesis (Fig. 2). These low proportions of photorespiration seem reasonable in *Trichodesmium*, which operates carbon concentrating mechanisms (44) to enhance carbon fixation and minimize the oxygenation reaction (i.e., photorespiration) by RuBisCO (14). After implementing $DPO_2$, modeled photorespiration rates were reduced by 42% and 35% in the dynamic-permeability case, respectively (Fig. 2, 4E, F and 5), resulting in a corresponding increase in gross fixed carbon allocated to biomass synthesis (Fig. 2). This was also found in the model experiments under constant light intensity (Fig. S4B). Further model experiments demonstrated that substituting the photorespiration rates in the dynamic-permeability case with those derived from the fixed-permeability case led to a reduction in *Trichodesmium* growth rates by 10% and 9% under low- and high-Fe conditions, respectively. These findings indicate that $DPO_2$ has the potential to improve carbon use efficiency and growth rate of *Trichodesmium* by reducing its photorespiration.

The reduction of photorespiration in the dynamic-permeability case (Fig. 4E and F) was partially attributed to the increased $O_2$ permeability during the early daytime (Fig. 3G, H, and 5), facilitating the diffusion of intracellular $O_2$ into the extracellular environment and resulting in lower intracellular $O_2$ concentration (Fig. 3E and F).

In addition, $DPO_2$ also resulted in a decrease in photosynthesis and $O_2$ production evolution during the early light period (Fig. 3A, B and 4A and B), contributing to reducing the intracellular $O_2$ concentration (Fig. 3E and F) and photorespiration rate (Fig. 4E, F, and 5). This can be attributed to the reduced need for respiratory protection during the

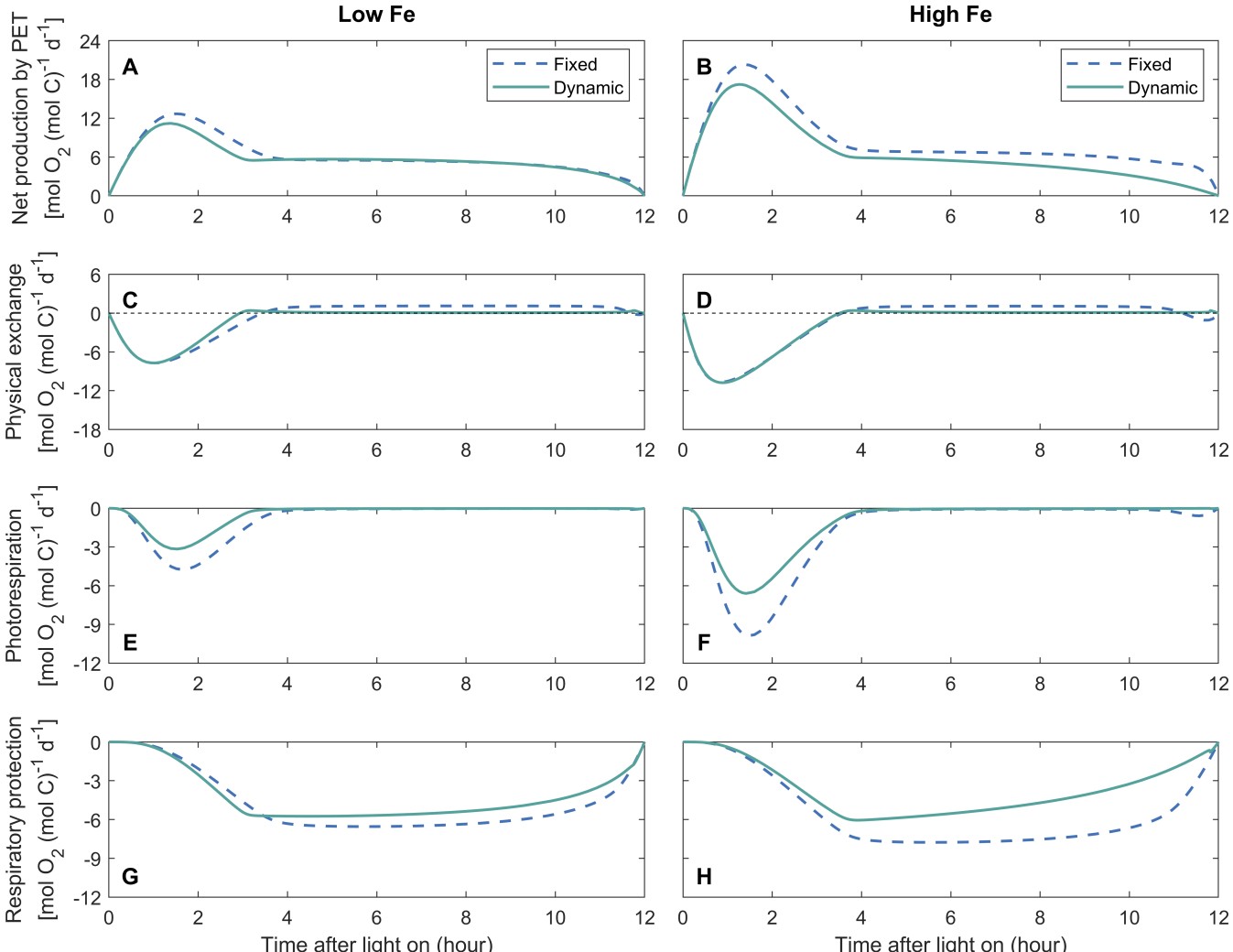

**FIG 4** Simulated diurnal intracellular $O_2$ fluxes. $O_2$ fluxes include net production by photosynthetic electron transfer (PET), physical exchange between intracellular and extracellular environments, $O_2$ consumption by photorespiration, and respiratory protection (negative values). Positive physical $O_2$ exchange represents $O_2$ flux into cells. The model is simulated with diurnally fixed or dynamic $O_2$ permeability of the cell membrane under low-Fe (40 pM) (A, C, E, and G) and high-Fe (1,250 pM) (B, D, F, and H) conditions.

low-$O_2$ window (Fig. 4G, H, and 5), thereby downregulating the requirement for carbon fixation (Fig. 3A, B , and 5) and Fe allocated to photosystems (Fig. S5A and B) during the early daytime (discussed later).

Laboratory experiments on *Trichodesmium* have shown that light-dependent $O_2$ uptake (which involves photorespiration as well as Mehler reaction and potentially flv-mediated $O_2$ uptake) can indeed be dynamically regulated by external $O_2$ and $CO_2$ concentrations: When external $O_2$ levels in the media were decreased or increased, light-dependent $O_2$ uptake changed proportionally within minutes in a reversible fashion (9). Another recent culture study of *Trichodesmium* also proposed a decrease in photorespiration under low-$O_2$ conditions (< 0.213 mol $O_2$ m$^{-3}$), although it was not quantified (16). In addition, the rate of light-dependent $O_2$ consumption has been shown to increase with an increase in $CO_2$ concentration in the medium (45).

A previous study proposed that photorespiration could serve as a mechanism to protect nitrogenase and $N_2$ fixation by consuming the $O_2$ produced by LPET, which, however, lacked support from observations or model simulations (46). In our model study, photorespiration primarily occurred during the early light period (Fig. 4E and F), when it was stimulated by high intracellular $O_2$ concentrations (Fig. 3E and F), while there

was minimal photorespiration during the low-$O_2$ window when $N_2$ fixation predominantly occurred (Fig. 4E and F). This low photorespiration could occur when high levels of respiratory protection depleted $O_2$ and produced $CO_2$ in *Trichodesmium* (Fig. 4G and H) (14), although the intracellular $CO_2$ concentration was not simulated in our study. In other words, this suggests that respiratory protection played a major role in lowering the intracellular $O_2$ level for $N_2$ fixation, while photorespiration may only have a limited contribution.

## Dynamic $O_2$ permeability reduces the requirement for respiratory protection

In the fixed-permeability case, 85% and 77% of the gross fixed carbon were allocated to respiratory protection to consume intracellular $O_2$ and create the low-$O_2$ window for $N_2$ fixation under low-Fe and high-Fe conditions, respectively (Fig. 2, 4G, and H). These percentages were generally consistent with previous model studies that adopted fixed $O_2$ permeability of the cell membrane over the light period (10, 12, 22) or even other $N_2$ fixers (47, 48). In comparison, in the dynamic-permeability case, the fractions of gross fixed carbon allocated to respiratory protection were reduced to 80% and 67% under low-Fe and high-Fe conditions, respectively (Fig. 2, 4G, and H). This reduction in respiratory protection was primarily attributed to the low-$O_2$ permeability in the dynamic-permeability case, which decreased the diffusion rate of extracellular $O_2$ into the cytoplasm (Fig. 4C, D, and 5). As a result, the intracellular stress on $N_2$ fixation caused by extracellular $O_2$ was basically relieved. Consequently, lower rates of $O_2$-consuming respiratory protection (Fig. 4G, H, and 5) were required to create and maintain lower intracellular $O_2$ concentrations (Fig. 3E and F and S4A, B), thereby supporting higher $N_2$ fixation rates in the dynamic-permeability case (Fig. 3A, B, and 5).

Previous model studies have proposed that respiratory protection is a crucial strategy for managing the intracellular $O_2$ level and creating the low-$O_2$ condition for $N_2$ fixation in *Trichodesmium* and other $N_2$-fixing cyanobacteria, such as *Crocosphaera* (4, 10, 48–50). Consistent with the previous study, our study demonstrated that respiratory protection respired major gross fixed carbon to consume intracellular $O_2$, resulting in a high indirect cost for $N_2$ fixation. This is also supported by observations of high daily-integrated gross fixed C:N ratios (e.g., references 30–50), even when Fe is replete (4, 9, 51, 52). Lowering carbon demand may allow this high ratio of carbon to be channeled into growth, thus increasing ecological competitiveness. Therefore, *Trichodesmium* might develop several strategies to reduce the requirement for respiratory protection and promote the carbon use efficiency, such as the dynamic Fe allocation, of which the function was quantified in a recent model study (22). On top of that, our study highlights the role of DPO$_2$ in lowering respiratory protection and alleviating the stress from $O_2$ on $N_2$ fixation in *Trichodesmium*.

## Dynamic $O_2$ permeability improves Fe use efficiency

The dynamic-permeability model case exhibited higher rates of $N_2$ fixation and growth compared to the fixed-permeability case under both low and high Fe levels (Fig. 2, 3 and 5 and S3), thereby promoting Fe use efficiency by 69% and 36%, respectively (Fig. 6).

The simulated diurnal patterns of Fe in both photosystems and nitrogenase suggest that DPO$_2$ could regulate intracellular Fe allocation (Fig. S5). Specifically, in the dynamic-permeability case, less Fe was allocated to photosystems during the light period compared to the fixed-permeability case (Fig. S5A and B). This can be attributed to two main reasons. First, the reduction in photorespiration allowed for the saving of energy and NADPH, which then can be utilized by carbon fixation. Second, the downregulation of respiratory protection decreased the demand for carbon fixation and storage in early daytime (Fig. 3 and 5). In addition, decreased respiratory protection mitigated the inhibition on photosynthetic electron transfer (4). Consequently, the Fe demand of photosystems was reduced in the dynamic-permeability case (Fig. 5 and S5A and B). The saved Fe from photosystems could then be allocated to (active) nitrogenase (Fig. 5 and S5C to F), enhancing $N_2$ fixation (Fig. 3C, D, and 5).

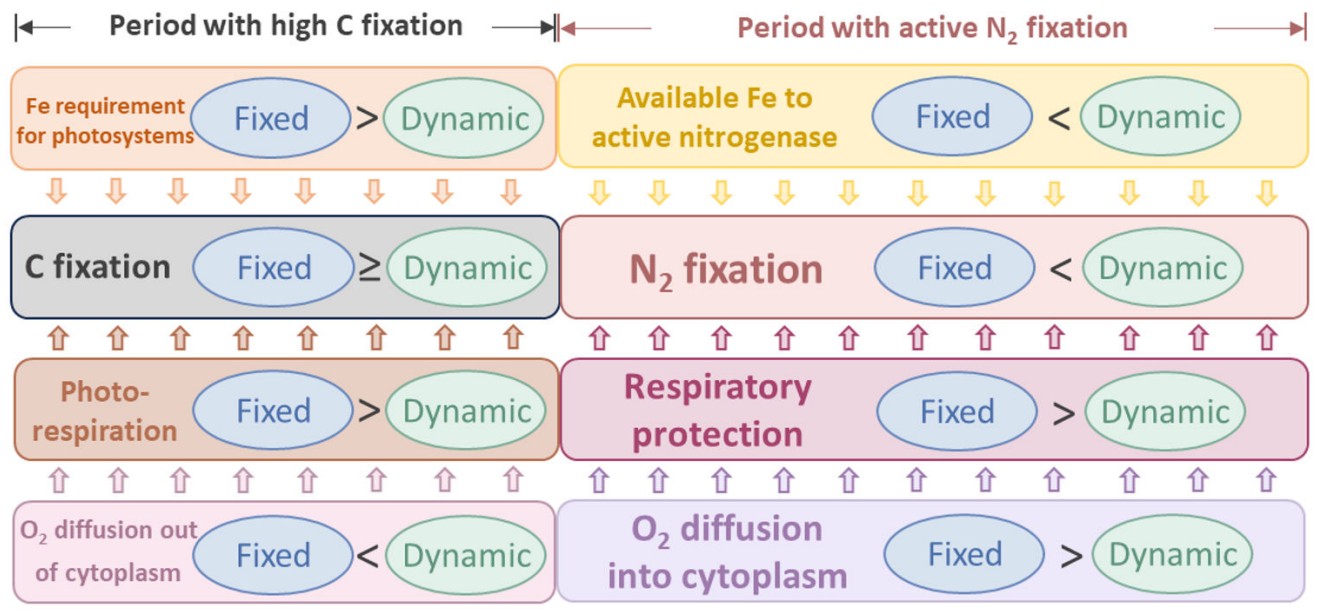

**FIG 5** Schematic diagram illustrating diurnally fixed and dynamic $O_2$ permeability of the cell membrane. Dynamic $O_2$ permeability can reduce photorespiration and the requirement for respiratory protection. In addition, it can promote carbon and iron use efficiency. Arrows between boxes represent the influence on carbon fixation and $N_2$ fixation.

In conclusion, our model study highlights two main potential mechanisms explaining the benefits of dynamic $O_2$ permeability to $N_2$ fixation and growth in *Trichodesmium*, which include the reduced photorespiration and decreased requirement for respiratory protection (Fig. 5). This can improve carbon and Fe use efficiency (Fig. 5 and 6), as well as $N_2$ fixation and growth rates of *Trichodesmium*, especially under low Fe. Therefore,

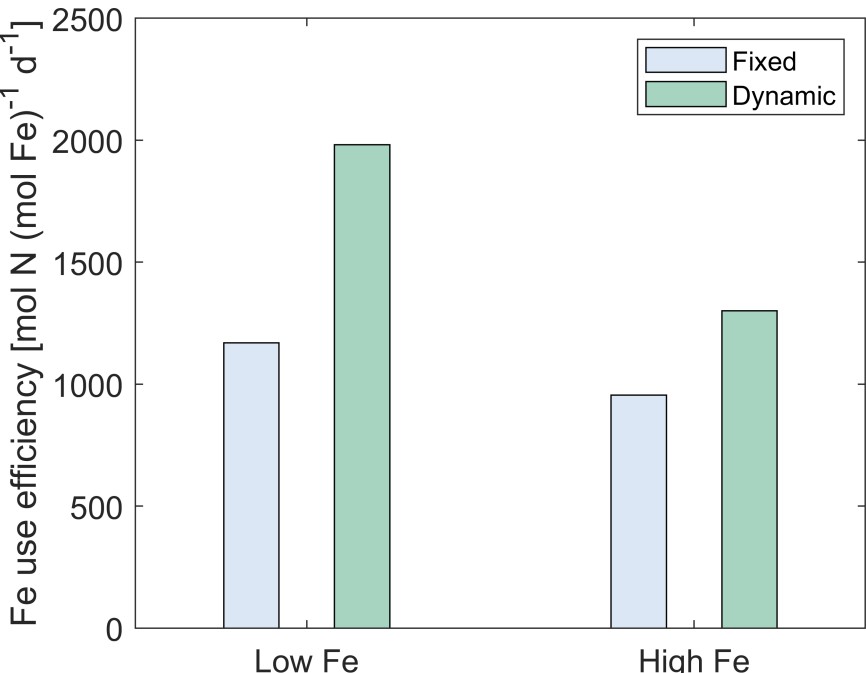

**FIG 6** Modeled results of Fe use efficiency in *Trichodesmium*. The model is simulated with diurnally fixed and dynamic $O_2$ permeability of the cell membrane under low-Fe (40 pM) and high-Fe (1,250 pM) conditions. Fe use efficiency: the ratio of the daily-integrated $N_2$ fixation rate to intracellular metabolic Fe.

this strategy potentially alleviates Fe limitation, helping *Trichodesmium* survive in the oligotrophic open ocean.

As phototrophic phytoplankton, *Trichodesmium* primarily inhabits the upper euphotic zone (53), where dissolved $O_2$ concentrations typically approach saturation (54). Baseline simulations were therefore set to the external $O_2$ concentration at 0.213 mol m$^{-3}$, which represents the saturation level for a typical upper ocean (salinity: 34 PSU; temperature: 25 °C) (29). Under specific circumstances, such as in the center of the microenvironment within *Trichodesmium* colonies, however, $O_2$ concentrations have been observed to be up to 200% saturation, especially under high light intensities (55). Model experiments across a range of $O_2$ concentrations (Fig. S6) show that photorespiration and nitrogenase inactivation become more intense and highlight that the benefits of dynamic $O_2$ permeability are amplified at elevated external $O_2$ levels. This suggests that dynamic permeability may be particularly important in *Trichodesmium* colonies at the sea surface subjected to high light intensities.

It should be noted that in our model, variations in the $O_2$ permeability of the *Trichodesmium* cell membrane were assumed to occur instantly in response to intracellular $O_2$ levels. However, further investigation is required to explore the timescale and the efficiency of redistributing hopanoid lipids in the cell membrane of *Trichodesmium*, which regulates the $O_2$ permeability. This will provide a more comprehensive understanding of the physiological benefits associated with DPO$_2$ in *Trichodesmium*. Further sensitivity model experiments of the maximal relative diffusion coefficient demonstrate that a proper range of variation/modulation in dynamic $O_2$ permeability is necessary for physiological benefits on $N_2$ fixation and growth in *Trichodesmium* (Fig. S7).

## Broader context: is dynamic $O_2$ permeability a common strategy in marine cyanobacterial diazotrophs?

The gene for the synthesis of hopanoids (the squalene-hopene cyclase gene, *shc*) was reported to be present in UCYN-A (nitroplast), UCYN-B (*Crocosphaera*), and UCYN-C (*Cyanothece*) (18, 56–61), suggesting their potential in dynamically regulating the membrane permeability to $O_2$ (Table 1). Hopanoids are known to decrease membrane $O_2$ permeability, protecting intracellular processes such as nitrogenase activity (18). However, direct evidence of diel variation in hopanoid production or dynamic changes in membrane lipid composition remains limited in marine diazotrophs.

A recent transcriptomic study (62) reveals diurnal cycling of gene expression in *Crocosphaera*, including genes related to photosynthesis and $N_2$ fixation, suggesting that membrane lipid composition may be dynamically regulated over diel cycles. Similarly, in non-diazotrophic cyanobacteria like *Synechocystis* sp. PCC 6803, diel light-dark cycles significantly influence fatty acid synthesis and membrane lipid turnover (63). Beyond cyanobacteria, studies in plants show that membrane lipid turnover can occur on timescales as short as 2 hours (64), suggesting that dynamic regulation of membrane lipids is a widespread strategy across diverse organisms. Together, these findings support the plausibility of DPO$_2$ as a physiological strategy in marine diazotrophs.

While the diel rhythms of photosynthesis and $N_2$ fixation in these marine non-heterocyst-forming diazotrophic cyanobacteria may differ from *Trichodesmium* (65, 66), it

**TABLE 1** Summary of indications from literature for dynamic $O_2$ permeability with potential benefits in some marine cyanobacterial diazotrophs

| Diazotroph | Indications/potential for dynamic $O_2$ permeability | References |
|---|---|---|
| *Trichodesmium* | Yes | This study and (18) |
| UCYN-A (nitroplast) | Possible | (58) |
| UCYN-B (*Crocosphaera*) | Yes | (56) |
| UCYN-C (*Cyanothece*) | Yes | (56) |
| Diazotroph-diatom association (DDA) | No | (18) |

is possible that the proposed physiological advantages of $DPO_2$ may apply to these diazotrophs.

The physiological roles of $DPO_2$ in improving the carbon and Fe use efficiency in some marine diazotrophs were similar to those of the dynamic Fe allocation (22). This suggests that dynamic regulation of membrane permeability can facilitate $N_2$ fixation and the growth of these diazotrophs in a manner similar to dynamic regulation of physiological processes, including temporal segregation (65, 67) and potential rapid mode switching (68) between photosynthesis and $N_2$ fixation. The in-depth controlling mechanisms of dynamic strategies guarantee further research.

## UCYN-A

The genome of UCYN-A (or nitroplast) (60) is highly reduced and lacks genes for $O_2$-producing photosystem II (69), suggesting a potential reduction in $O_2$-induced stresses and photorespiration. A previous study demonstrated coordination between the expression of the shc gene and the *nifH* gene (encoding nitrogenase) in UCYN-A (56). This indicates that UCYN-A may regulate membrane permeability to $O_2$ dynamically, lowering intracellular $O_2$ levels and protecting nitrogenase activity by mitigating $O_2$ diffusion from the $O_2$-producing haptophyte algal host cell and the ambient environment into the cytoplasm during active $N_2$ fixation. In addition, the low transcription level of the cytochrome c oxidase *coxA* gene in UCYN-A during the light period (26, 56) indicates a reduced requirement for respiratory protection. According to insights from this study, such dynamic regulation could enhance carbon and Fe use efficiency in UCYN-A.

## UCYN-B and UCYN-C

Both UCYN-B (*Crocosphaera*) and UCYN-C (*Cyanothece*) conduct photosynthesis during the light period and $N_2$ fixation at night (56, 70–72). Transcriptomic analysis has shown that the shc gene in UCYN-B and UCYN-C reached its peak expression just before the increase in the nifH gene expression level (56). This observation suggests that UCYN-B and UCYN-C may produce hopanoids to protect $N_2$ fixation from $O_2$ diffusing from the external environment during the dark period, therefore reducing the level of respiratory protection required to safeguard nitrogenase and support $N_2$ fixation (48, 49). In addition, lower expression of the shc gene during the daytime in UCYN-B and UCYN-C (56) might result in higher $O_2$ permeability, facilitating the diffusion of photosynthetically produced $O_2$ out of the cell and thus reducing photorespiration. These mechanisms, as proposed in this study, would lead to elevated carbon and Fe use efficiency. In addition, these organisms have heterogeneous rates of $N_2$ fixation (i.e., some cells fix $N_2$ and others do not) (73), and thus, the most effective $DPO_2$ would also be heterogeneous across their population. Furthermore, the presence of hopanoid rafts in UCYN-B (74) suggests that hopanoid may be redistributed within the membrane to dynamically modulate $O_2$ permeability (20).

## DDAs

Heterocyst-forming diazotrophs (diatom-diazotroph assemblage, DDA) lack the shc gene responsible for hopanoid synthesis (18). This indicates that $DPO_2$ may not be necessary in DDA. Instead, the heterocyst uses glycolipids to form a barrier against extracellular $O_2$ (75, 76), seemingly providing sufficient protection for nitrogenase activity and reducing Fe requirements compared to non-heterocyst-forming cyanobacterial diazotrophs (77). Similarly, the absence of the shc gene in non-diazotrophic cyanobacteria such as *Prochlorococcus* and *Synechococcus* (18) further supports the idea that $DPO_2$ is one of the evolved strategies of diazotrophs when facing $O_2$ stress on $N_2$ fixation.

## Conclusions

We investigated how dynamic cell permeability to $O_2$ ($DPO_2$) in *Trichodesmium* trichomes may enhance $N_2$ fixation and growth rates, taking into account the effect of photorespiration. Our model shows that $DPO_2$ reduces photorespiration, especially during the early daytime, lowering the requirement for respiratory protection, facilitating the formation of the low-$O_2$ intracellular condition for $N_2$ fixation, and improving carbon use efficiency. Moreover, $DPO_2$ in *Trichodesmium* may impact the diurnal allocation of the intracellular Fe, ultimately promoting the Fe use efficiency. Fragmental evidence for $DPO_2$ is reported in other marine diazotrophs, suggesting that $DPO_2$ is a common strategy adopted by marine diazotrophs. The model framework presented in our study could also be used to explore other physiological mechanisms that control $N_2$ fixation, such as light and Fe colimitation. It can also be incorporated into biogeochemical models to enhance their predictive capabilities in the ecophysiology of marine diazotrophs.

### ACKNOWLEDGMENTS

This project is partly supported by the National Natural Science Foundation of China (42076153 and 42376140 to YWL), China Scholarship Council, and MEL PhD Fellowship to WL. This work was supported by a grant from the Simons Foundation (LS-ECIA-MEE-00001549, Inomura). O.P. is supported by GACR 23-06593S and by OP JAK Photomachines. M.E. is supported by GACR GA24-12396S and by OP JAK project Photomachines.

W.L. and Y.W.L. originated the concept for the study. W.L. designed the numerical model. W.L. coded the initial version of the model and performed numerical modeling. W.L., K.I., O.P., M.E., and Y.W.L. analyzed the results and improved the numerical model. W.L. wrote the first draft of the manuscript, and all coauthors revised the manuscript.

### AUTHOR AFFILIATIONS

[1]State Key Laboratory of Marine Environmental Science and College of Ocean and Earth Sciences, Xiamen University, Xiamen, China
[2]Centre Algatech, Institute of Microbiology of the Czech Academy of Sciences, Třeboň, Czech Republic
[3]Institute for Advanced Study, Shenzhen University, Shenzhen, China
[4]Graduate School of Oceanography, University of Rhode Island, Narragansett, Rhode Island, USA

### AUTHOR ORCIDs

Weicheng Luo http://orcid.org/0000-0002-4377-2030
Keisuke Inomura http://orcid.org/0000-0001-9232-7032
Ondřej Prášil http://orcid.org/0000-0002-0012-4359
Meri Eichner http://orcid.org/0000-0001-6106-7880
Ya-Wei Luo http://orcid.org/0000-0001-6106-7901

### AUTHOR CONTRIBUTIONS

Weicheng Luo, Conceptualization, Data curation, Formal analysis, Investigation, Methodology, Software, Validation, Visualization, Writing – original draft, Writing – review and editing | Keisuke Inomura, Formal analysis, Funding acquisition, Methodology, Validation, Visualization, Writing – review and editing | Ondřej Prášil, Formal analysis, Funding acquisition, Methodology, Validation, Visualization, Writing – review and editing | Meri Eichner, Formal analysis, Funding acquisition, Methodology, Validation, Visualization, Writing – review and editing | Ya-Wei Luo, Conceptualization, Formal analysis, Funding acquisition, Methodology, Validation, Visualization, Writing – review and editing

## DATA AVAILABILITY

All data, code, and materials used in this study are available from the corresponding author upon reasonable request. The code is freely available in figshare (https://doi.org/10.6084/m9.figshare.28829270).

## ADDITIONAL FILES

The following material is available online.

### Supplemental Material

**Supplemental material (Spectrum00453-25-S0001.pdf).** Supplemental methods, Tables S1 to S4, and Fig. S1 to S7.

### Open Peer Review

**PEER REVIEW HISTORY (review-history.pdf).** An accounting of the reviewer comments and feedback.

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
