## [Reviewer comments · Microbiology Spectrum]

Microbiology Spectrum

Modeling dynamic oxygen permeability as a mechanism to mitigate oxygen-induced stresses on photosynthesis and N₂ fixation in marine *Trichodesmium*

Weicheng Luo, Keisuke Inomura, Ondřej Prášil, Meri Eichner, and Ya-Wei Luo

Corresponding Author(s): Weicheng Luo, Xiamen University

Review Timeline:

Submission Date:	February 24, 2025
Editorial Decision:	March 22, 2025
Revision Received:	May 12, 2025
Editorial Decision:	June 5, 2025
Revision Received:	June 10, 2025
Accepted:	June 23, 2025

Editor: Hongan Long

Reviewer(s): Disclosure of reviewer identity is with reference to reviewer comments included in decision letter(s). The following individuals involved in review of your submission have agreed to reveal their identity: Nir Keren (Reviewer #1)

Transaction Report:

DOI: <https://doi.org/10.1128/spectrum.00453-25>

Re: Spectrum00453-25 (**Modeling dynamic oxygen permeability as a mechanism to mitigate oxygen-induced stresses on photosynthesis and N₂ fixation in marine *Trichodesmium***)

Dear Mr. Weicheng Luo:

Thank you for the privilege of reviewing your work. Below you will find my comments, instructions from the Spectrum editorial office, and the reviewer comments.

Revision Guidelines

Sincerely,
Hongan Long
Editor
Microbiology Spectrum

Reviewer #1 (Comments for the Author):

The manuscript "Modeling dynamic oxygen permeability as a mechanism to mitigate oxygen-induced stresses on photosynthesis and N₂ fixation in marine *Trichodesmium*" By Luo and coworkers is a theoretical study examining the potential of dynamic changes in hopanoid concentration in regulating intracellular oxygen concentrations. The authors demonstrate the potential benefits of these dynamic changes for both photosynthesis and nitrogen fixation processes. In my review I will focus my comments on the physiological aspects. I am not an expert on modelling. My specific comments are included below.

Highlights: I would appreciate a bit more clearly - how DPO2 affects these processes.

Line 66: photorespiration should not appear in "". Photorespiration also wastes reduced nitrogen. This should be mentioned.

Line 90: please expand on the molecular mechanisms by which hopanoids effect membrane permeability.

Line 92: are there experimental evidence for the dynamics of hopanoid production, or is it strictly a hypothesis? Please clarify this.

Figure 1: The model simulates physiological processes in *Trichodesmium trichome*. Does the model take into account heterogeneity of cells along the trichome?

Line 193: Why is Fe considered and not other relevant metals. Should Mo also be included in the model? Was the speciation of Fe considered? The dissolved Fe pool is very small regardless of its concentration.

Line 225: Why did the simulations encompass ten Fe levels? This needs to be explained.

Figure 2 and its discussion: Does dynamic O2 permeability have fixed rates in the model. Are their variables to this process that should be explored (as in modifying equation 1)?

Figures 2-4: This may be a result of my lack of knowledge in modeling - are the models deterministic? Do running them several times will always result in the same outcome? I am asking because I do not see repeats or error values in the results figures.

Reviewer #2 (Public repository details (Required)):

- 1 Data for selecting model parameters
- 2 All simulation data for model evaluation
- 3 Programming code for implementing the model and selecting parameters

See detailed comments in the attachment.

Dear Editor and Authors,

Thank you for the opportunity to review this manuscript. The study addresses an interesting and relevant topic concerning dynamic oxygen permeability and its impact on *Trichodesmium*. The research is well-motivated and has the potential to contribute to our understanding of how N₂-fixing microorganisms optimize their physiological processes in response to environmental challenges.

However, I am not an expert in physiology. Nonetheless, I have identified several methodological and data simulation issues that need to be addressed.

Major :

1. As I understand it, the model presented in this manuscript is a modification of the model used in the paper titled "*Diurnally dynamic iron allocation promotes N₂ fixation in the marine dominant diazotroph Trichodesmium*", with the key difference being a focus on oxygen permeability rather than iron dynamics. However, based on the results shown in Figures 2, 3, 4, S2, S3, and S5, one of the control variables in this study remains low Fe and high Fe, which are identical to those in the previous study.

Given that this study aims to investigate the effects of fixed and dynamic O₂ permeability, it might be more appropriate to adjust the simulation settings to compare low O₂ and high O₂ conditions instead of Fe availability. Such a modification could provide more direct insights into how oxygen permeability affects *Trichodesmium* physiology.

For instance:

Figure S3 could present the simulated daily-integrated N₂ fixation rates, growth rates, and their relative changes between “Dynamic permeability” and “Fixed permeability” cases under different O₂ concentrations.

Similarly, the data in Figure 2 could be simulated under diurnally fixed and dynamic O₂ permeability conditions with low-O₂ and high-O₂ scenarios.

2. My second concern relates to the selection of hyperparameters (i.e., fixed model parameters in this study). For a mathematical model with multiple variables, the model's performance can be highly sensitive to the choice of these hyperparameters. In this study, some parameter values were adopted from previous literature, while others were derived from laboratory culture data (Table S2). It would be helpful if the authors could provide more details on how these parameter values were selected. Specifically: 1) Were these parameters obtained under

consistent environmental conditions? 2) How well do these selected values correspond to natural *Trichodesmium* populations? Providing additional justification for parameter selection would enhance the credibility of the model and its applicability.

For instance:

Line 142: The study compares the new dynamic O₂ permeability (ϵ) with a fixed ϵ of 1.0×10^{-4} . Could the authors clarify why this specific value was chosen? Would the results change significantly if a different reference value were used?

Line 160: The manuscript states that $k_{O_2}^{PR} = 1.92$ is the half-saturation coefficient of O₂ for photorespiration (Fig. S1B). However, based on Fig. S1B, the value of $k_{O_2}^{PR}$ seems to range between 0.8 and 1.2. Could the authors clarify how this specific value was determined?

Given the number of fixed parameters used in the model, I would encourage the authors to carefully verify all parameter values to ensure they are consistent with the model's assumptions and relevant environmental conditions.

3. Following the concerns in #2, as a methodological paper, the model was validated using simulation data with multiple hyperparameters. However, since the simulation dataset is not publicly available and the manuscript lacks sufficient details regarding the simulation data, I am concerned about the possibility of overfitting. Given that a larger number of coefficients generally increases the risk of overfitting, further clarification on this aspect would be beneficial.

Also, it is unclear how many simulation runs were performed for each setting, as this information is not mentioned in the manuscript. If only a single simulation was conducted per setting, the results would essentially represent a single-point estimate, which may lack statistical significance for model validation. I would suggest running multiple simulations under the same conditions and reporting either the average values or confidence intervals to improve the robustness of the results. In addition, incorporating appropriate statistical tests would be highly recommended to verify the model's performance and ensure the reliability of the findings.

Furthermore, the more critical aspect is that the model should ideally be validated against actual experimental data rather than relying solely on simulations. Providing such validation would significantly strengthen the credibility of the study.

4. When adding more variables to a model, it is essential to consider the correlation between them. Another key concern regarding this model is the lack of consideration for variable correlation. In complex environmental and physiological models, multiple parameters can be interdependent, and failing to account for these relationships may introduce multicollinearity issues, reduce model stability, and affect the reliability of the results.

For instance:

Dynamic Fe availability and O₂ permeability both impact N₂ fixation, along with the newly introduced variables photorespiration (PR) and respiratory protection (RP). The correlation between these variables should be tested and accounted for to ensure that the model does not overfit to highly dependent variables.

Equation (4) in the manuscript represents the rate regulation by two substrates (O₂ and CH₂O), but it does not account for their interaction. In reality, these substrates clearly influence each other, and incorporating their interaction into the model would enhance its accuracy and robustness

5. Ensuring mathematical rigor and appropriate statistical validation is crucial in any modeling study. In the current manuscript, there are several issues related to both the statistical evaluation of results and the clarity of mathematical notation, which should be addressed to strengthen the study's reliability.

Statistical Validation and Justification of Conclusions: In Figure S2, the reported $R^2=0.12$ suggests a weak correlation between the model predictions and observed data. It may be beneficial to provide additional statistical assessments to further support the model's validity. If the low R^2 value is due to natural variability in the data, a brief discussion on why the model remains reliable despite the weak correlation would strengthen the interpretation of the results.

Consistency in Mathematical Notation and Symbol Definition: Throughout the manuscript, many mathematical symbols are introduced before they are properly defined, making it difficult to follow the derivations and model descriptions.

Line 134: The O₂ permeability is introduced, but it is formally defined as ϵ later in the text.

Line 187: The CCM is short for carbon concentrating mechanisms should be mentioned in the manuscript.

Minor:

1. Ensure that all figures use consistent axis labeling and font sizes for readability.
2. The manuscript frequently refers to "fixed O₂ permeability case" and "dynamic permeability case" but does not clearly define these terms early in the text. Better to add a brief definition of these cases in the Introduction or Methods section
3. In Figure S2, it would be helpful to explain the significance of error bar.
- 4.

Main text

Line 206: "including" → "These parameters include..."

Line 107: "model schemes" → "model structure"

Line 299 " seem to be reasonable" → " seem reasonable" , "which operate" → "which operates"

Line 350 "into cytoplasm" → "into the cytoplasm"

Line 384 "including" → "which include"

Line 382 "Another previous study" → "Another study" or "A previous study"

Supplementary

Line 104: "basically normalized" → "which are normalized to carbon biomass"

Line 157: "for simplification" → "for simplicity"

Line 271: "The model is simulated" → "The model simulations were performed"

We have taken all the comments of the reviewers into account in the revision; replies to each of the comments are provided below in blue fonts. Please note that all the line numbers mentioned in the response refer to the Marked-up Manuscript.

Reviewer comments:

Reviewer #1 (Comments for the Author):

The manuscript "Modeling dynamic oxygen permeability as a mechanism to mitigate oxygen-induced stresses on photosynthesis and N₂ fixation in marine *Trichodesmium*" By Luo and coworkers is a theoretical study examining the potential of dynamic changes in hopanoid concentration in regulating intracellular oxygen concentrations. The authors demonstrate the potential benefits of these dynamic changes for both photosynthesis and nitrogen fixation processes. In my review I will focus my comments on the physiological aspects. I am not an expert on modelling. My specific comments are included below.

Response:

We thank the reviewer for the time and the thoughtful review of our manuscript. We appreciate the reviewer's insights, particularly regarding the physiological aspects of our study. The reviewer's feedback has helped us refine and strengthen our work. We have carefully addressed the reviewer's specific comments and made the necessary clarifications to improve the clarity and scientific rigor of the manuscript.

Highlights: I would appreciate a bit more clearly - how DPO₂ affects these processes.

Response:

We have expanded the "Highlights" section to explicitly describe the mechanism by which dynamic oxygen permeability (DPO₂) increases N₂ fixation and growth rates. Please refer to Highlights:

"DPO₂ increases N₂ fixation and growth rates by reducing O₂-induced stress and increasing the amount of active nitrogenase during low-O₂ window." (Lines 16 and 17).

Line 66: photorespiration should not appear in "". Photorespiration also wastes reduced nitrogen. This should be mentioned.

Response:

We have removed the quotation marks around photorespiration as suggested. Additionally, we have explicitly mentioned that photorespiration consumes reduced nitrogen in the "Introduction" section of the revised manuscript. (Lines 67 and 69).

Line 90: please expand on the molecular mechanisms by which hopanoids effect membrane permeability.

Response:

We have introduced the molecular mechanisms by which hopanoids effect membrane permeability in the revised manuscript:

“*Trichodesmium* can synthesize hopanoids, which can be intercalated into lipid bilayers of membranes (Cornejo-Castillo and Zehr, 2019; Poger and Mark, 2013). The planar and hydrophobic structure of hopanoids may decrease the membrane permeability to O₂. Also, hopanoids may form rafts (high concentration domain), which can be distributed within the membrane, and thus may dynamically regulate cell permeability to O₂ (dynamic-permeability model case) (Belin et al., 2018; Cornejo-Castillo and Zehr, 2019).” (Lines 91 to 96).

Line 92: are there experimental evidence for the dynamics of hopanoid production, or is it strictly a hypothesis? Please clarify this.

Response:

We appreciate the reviewer’s comment. We believe that it is currently a hypothesis. Current evidence is solely based on the set of genes that may allow hopanoid synthesis. To provide evidence for the dynamic regulation of hopanoid production, transcriptomics and/or measurements of hopanoids for various time points need to be done, but we are not aware of such publications. At the same time, *Trichodesmium* has distinct diurnal cycle, and highly dynamic protein regulations (Frischkorn et al., 2018), thus, it is highly likely that similar dynamics may apply hopanoid production. To convey this point and clarify the hypothetical aspect we revised the sentence.

“Such a dynamic regulation is currently a hypothesis, it is likely that the dynamic expression of hopanoid biosynthesis genes may occur, given the highly dynamic protein expression in *Trichodesmium* (Frischkorn et al., 2018).” (Lines 96 to 98).

Figure 1: The model simulates physiological processes in *Trichodesmium* trichome. Does the model take into account heterogeneity of cells along the trichome?

Response:

We thank the reviewer for the comments. Our current model does not account for the heterogeneity of cells along the trichome. Instead, we assume that all cells share the same physiological state, as assumed in our previous model, which demonstrated that spatial segregation of N₂ fixation and photosynthesis into different cell segments may not be required for *Trichodesmium* trichome (Luo et al., 2022). This assumption is based on the evidence that most cells are capable of both N₂ fixation and photosynthesis (Ohki, 2008).

Integrating cell heterogeneity into the model and investigating its potential role in regulating N₂ fixation and growth in *Trichodesmium* would be a valuable direction for future research.

Line 193: Why is Fe considered and not other relevant metals. Should Mo also be included in the model? Was the speciation of Fe considered? The dissolved Fe pool is very small regardless of its concentration.

Response:

We thank the reviewer for thoughtful comment. Regarding Mo and/or Fe, we would like to clarify them as below:

- (1) The primary reason is that in the ocean, Fe is the key trace metal that limits the phytoplankton in the ocean. There has been growing evidence of Fe limitation of N₂ fixation/photosynthesis in *Trichodesmium* (Berman-Frank et al., 2001; Chappell et al., 2012; Kupper et al., 2008; Shi et al., 2007; Zehr and Capone, 2020), and thus, we targeted the inclusion of Fe for the present model development.
- (2) Mo is indeed a critical component of nitrogenase; however, due to limited observational data, its role in regulating *Trichodesmium* N₂ fixation under both sufficient and limiting conditions remains unclear. The lack of such data constrains our ability to incorporate Mo dynamics into the model with confidence. Also, Mo seems not a limiting factor for the growth of marine phytoplankton (Howarth and Cole, 1985) including *Trichodesmium*; thus, we consider its effect secondary in the ocean.
- (3) Fe was explicitly considered in our model due to its well-documented role in N₂ fixation and photosynthesis, as well as the availability of experimental data that help constrain Fe-related physiological processes (Shi et al., 2012).
- (4) Regarding Fe speciation, our model currently does not resolve explicit Fe chemical forms. Previous work demonstrated that there is a strong relationship between the Fe quota in *Trichodesmium* cells and the dissolved Fe concentration in sea water (Shi et al., 2012). Hence, the dissolved Fe concentration used as the model input implicitly represents Fe availability as a limiting factor for *Trichodesmium*. Future work could incorporate Fe speciation dynamics to better capture its bioavailability, especially given the typically low dissolved Fe concentrations in marine environments (Tagliabue et al., 2016).

Line 225: Why did the simulations encompass ten Fe levels? This needs to be explained.

Response:

We are thankful to the reviewer for the comment. The simulations encompassed ten Fe levels to explore the benefits of dynamic diffusion within the observed range of *Trichodesmium* Fe quota.

We have also revised the manuscript as below (Lines 236 and 239):

“Similar to a previous model study (Luo and Luo, 2023), the simulations encompassed ten Fe' levels (20 pM to 1800 pM) to approximately represent *Trichodesmium* Fe quota of 10 to 1000 μmol Fe

(mol C)⁻¹, a range observed in field *Trichodesmium* samples (Luo et al., 2019).”.

Figure 2 and its discussion: Does dynamic O₂ permeability have fixed rates in the model. Are their variables to this process that should be explored (as in modifying equation 1)?

Response:

We appreciate the reviewer’s comments. In the model, dynamic O₂ permeability does involve some fixed parameters related to diffusion rates of O₂, including ϵ_{max} (the maximal relative diffusion coefficient; 2.0×10^{-4}) and $k_{O_2}^{diff}$ (the half-saturation constant of O₂ for the relative diffusivity to seawater; $0.213 \text{ mol O}_2 \text{ m}^{-3}$) (Equation 1). These parameters are constrained by observational data (Shi et al., 2012) assuming the existence of dynamic O₂ permeability, thus we do not vary these parameters in the standard version of our model.

Per the reviewer's suggestion, we performed further sensitivity model experiments of ϵ_{max} . The results suggest that the proper variation of dynamic O₂ permeability would allow for physiological benefits on N₂ fixation and growth in *Trichodesmium*, with the current value of ϵ_{max} (100% in New Figure S7) giving approximately the highest benefits. We also added discussion accordingly (lines 412 to 414):

“Further sensitivity model experiments of the maximal relative diffusion coefficient demonstrate that a proper range of variation/modulation in dynamic O₂ permeability is necessary for physiological benefits on N₂ fixation and growth in *Trichodesmium* (Fig. S7).”.

FIG S7 Simulated relative changes of daily-integrated N₂ fixation rates and growth rates between “Dynamic permeability” and “Fixed permeability” cases under different maximal relative diffusion coefficient (ϵ_{max}), from 25% to 400% of that in the standard case (Fig. 1B). The model simulations were performed under low-Fe (40 pM) and high-Fe (1250 pM) conditions.

Figures 2-4: This may be a result of my lack of knowledge in modeling - are the models deterministic? Do running them several times will always result in the same outcome? I am asking because I do not see repeats or error values in the results figures.

Response:

We thank the reviewer for the question. The model is deterministic, meaning that the results are derived based on the principle of optimizing the intracellular resource allocations (e.g., carbon, iron and energy) for maximal growth. As a result, running the optimization module of our model multiple times will always yield the same best set of parameters with the optimal outcome. This is why error bars are not included in the results figures.

Reviewer #2 Public repository details (Required) and (Comments for the Author)

- 1 Data for selecting model parameters
- 2 All simulation data for model evaluation
- 3 Programming code for implementing the model and selecting parameters

Response:

We thank the reviewer for these suggestions. We appreciate insightful comments of the reviewer, which have helped improve the transparency and reproducibility of our work.

- (1) Data for selecting model parameters – The data used for parameter selection has been made available for reference.
- (2) All simulation data for model evaluation – We have deposited all relevant simulation data in the open-access repository figshare.
- (3) Programming code for implementing the model and selecting parameters – The model code, along with scripts for parameter selection and implementation, has also been uploaded to figshare.

The dataset and code can be accessed at the following free link:
<https://doi.org/10.6084/m9.figshare.28829270>.

Dear Editor and Authors,

Thank you for the opportunity to review this manuscript. The study addresses an interesting and relevant topic concerning dynamic oxygen permeability and its impact on *Trichodesmium*. The research is well-motivated and has the potential to contribute to our understanding of how N₂-fixing microorganisms optimize their physiological processes in response to environmental challenges. However, I am not an expert in physiology. Nonetheless, I have identified several methodological and data simulation issues that need to be addressed.

Response:

We are thankful to the reviewer for taking the time to review our manuscript and for the constructive feedback. We appreciate positive remarks from the reviewer on the relevance and potential contributions of our study.

We also acknowledge the reviewer's concerns regarding methodological and data simulation aspects. We have carefully and hopefully addressed the identified issues in our revision.

Major:

1. As I understand it, the model presented in this manuscript is a modification of the model used in the paper titled "*Diurnally dynamic iron allocation promotes N₂ fixation in the marine dominant diazotroph Trichodesmium*", with the key difference being a focus on oxygen permeability rather than iron dynamics. However, based on the results shown in Figures 2, 3, 4, S2, S3, and S5, one of the control variables in this study remains low Fe and high Fe, which are identical to those in the previous study.

Given that this study aims to investigate the effects of fixed and dynamic O₂ permeability, it might be more appropriate to adjust the simulation settings to compare low O₂ and high O₂ conditions instead of Fe availability. Such a modification could provide more direct insights into how oxygen permeability affects *Trichodesmium* physiology.

For instance:

Figure S3 could present the simulated daily-integrated N₂ fixation rates, growth rates, and their relative changes between “Dynamic permeability” and “Fixed permeability” cases under different O₂ concentrations.

Similarly, the data in Figure 2 could be simulated under diurnally fixed and dynamic O₂ permeability conditions with low-O₂ and high-O₂ scenarios.

Response:

We are grateful for the reviewer’s thoughtful and insightful comments. We appreciate the reviewer’s suggestion to explicitly compare low- and high-O₂ conditions, as it highlights an important dimension of O₂ permeability dynamics in *Trichodesmium*.

The reviewer is correct that our study builds upon our previous iron-focused model (Luo and Luo, 2023), but here we prioritize O₂ permeability as the central innovation. The retention of Fe availability as a control variable reflects two key rationales:

- (1) Ecological relevance: Collected data from field observations (Luo et al., 2019) (Please refer to Figure S3 in this reference) demonstrate that *Trichodesmium* inhabits marine environments across a broad range of iron concentrations (up to about 2000 pM for seawater inorganic iron concentration), and its iron use efficiency differs between iron-limited and iron-replete conditions (evidenced in Figure 6 of this study and Table 1 of Luo et al., 2019). Our simulations capture this realistic physiological context, wherein dynamic O₂ permeability may enhance Fe use efficiency—a potential adaptive benefit.
- (2) Metabolic coupling: Fe could modulate cellular energy production by Fe-requiring photosynthetic electron transfer, photosynthesis rates and also nitrogenase protection mechanisms, both of which are tightly linked to intracellular O₂ management. Thus, varying Fe

conditions inherently test O₂ permeability's role under realistic metabolic constraints.

We acknowledge the reviewer's suggestion about O₂ gradients. *Trichodesmium* is phototrophic phytoplankton lives in euphotic zones, where ambient O₂ levels are typically near saturation based on data from World Ocean Atlas (WOA) or Global Ocean Data Analysis Project (GLODAP). There could be microscale environment within colonies (e.g., in the center) can exhibit, for example, hyperoxia (e.g., 200% saturation) (Eichner et al., 2017). To address this, we have conducted the following revisions:

- (1) Discussion: Expand the interpretation of how dynamic permeability could mitigate stress from O₂ under different O₂ levels (Lines 404 to 406):
“These mechanisms could also explain model outcomes across varying O₂ concentrations (Fig. S6), where higher O₂ levels increase photorespiration and intensify nitrogenase inactivation, with dynamic O₂ permeability demonstrating amplified benefits in mitigating these constraints.”.
- (2) Supplementary Information: Include new figures (i.e., Figure S6) comparing N₂ fixation and growth rates under fixed vs. dynamic permeability across a gradient of O₂ concentrations (low/high/saturation/hyperoxia).

We hope these revisions will clarify why ambient O₂ variability was not our primary focus while still addressing its localized ecological significance. We are happy to refine this further based on the reviewer's feedback. We thank the reviewer again for the valuable input, which has strengthened our manuscript.

FIG S6 Simulated daily-integrated N₂ fixation rates (A, B), growth rates (C, D) and their relative changes (E, F) between “Dynamic permeability” and “Fixed permeability” cases under different ambient oxygen concentrations, from 25% to 200% of the saturation. The model simulations were performed under low-Fe (40 pM) and high-Fe (1250 pM) conditions.

2. My second concern relates to the selection of hyperparameters (i.e., fixed model parameters in this study). For a mathematical model with multiple variables, the model's performance can be highly sensitive to the choice of these hyperparameters. In this study, some parameter values were adopted from previous literature, while others were derived from laboratory culture data (Table S2). It would be helpful if the authors could provide more details on how these parameter values were selected. Specifically: 1) Were these parameters obtained under consistent environmental conditions? 2) How well do these selected values correspond to natural *Trichodesmium* populations? Providing additional justification for parameter selection would enhance the credibility of the model and its applicability.

Response:

We are grateful to the reviewer for insightful suggestions regarding hyperparameter selection. Below we provide a point-by-point clarification to address these concerns:

(1) Parameter Sources & Consistency:

- Fixed stoichiometric or morphological parameters were adopted from established literature, ensuring alignment with fundamental biochemical or morphological constraints as below:

Symbol	Unit	Definition	Value	Source or note
Elemental or energy stoichiometries of metabolic activities				
q_{LPET}^{NADPH}	mol NADPH (mol electron) ⁻¹	NADPH/electron ratio of LPET	0.5	(Allen, 2003)
q_{LPET}^{ATP}	mol ATP (mol electron) ⁻¹	ATP/electron ratio of LPET	0.65	(Geider et al., 2009)
$q_{LPET}^{O_2}$	mol O ₂ (mol electron) ⁻¹	O ₂ /electron ratio of LPET	0.25	(Allen, 2003)
q_{AET}^{ATP}	mol ATP (mol electron) ⁻¹	ATP/electron ratio of MR-AET	0.65	(Geider et al., 2009)
q_{NF}^{NADPH}	mol NADPH (mol N) ⁻¹	NADPH/N ratio of N ₂ fixation	3	Flores and Herrero, 1994)
q_{NF}^{ATP}	mol ATP (mol N) ⁻¹	ATP/N ratio of N ₂ fixation	9	(Flores et al., 2005; Flores and Herrero, 1994)
q_{PR}^{NADPH}	mol NADPH (mol C) ⁻¹	NADPH/C ratio of photorespiration	4	(Bauwe et al., 2010)
q_{PR}^{ATP}	mol ATP (mol C) ⁻¹	ATP/C ratio of photorespiration	7	(Bauwe et al., 2010)
$q_{C,PR}^{O_2}$	mol O ₂ (mol C) ⁻¹	O ₂ /C ratio of photorespiration	3	(Bauwe et al., 2010)
q_{CF}^{NADPH}	mol NADPH (mol C) ⁻¹	NADPH/C ratio of C fixation	2	(Baker et al., 2007)
q_{CF}^{ATP}	mol ATP (mol C) ⁻¹	ATP/C ratio of C fixation	3	(Baker et al., 2007)
q_{CCM}^{ATP}	mol ATP (mol C) ⁻¹	ATP/C ratio of CCM	0.8	(Raven et al., 2014)
q_{BIO}^{ATP}	mol ATP (mol C) ⁻¹	ATP/C ratio of biosynthesis	2	(Inomura et al., 2019)
q_{RESP}^{ATP}	mol ATP (mol C) ⁻¹	ATP/C ratio of respiration	5	(Mitchell, 1970)
$q_C^{O_2}$	mol O ₂ (mol C) ⁻¹	O ₂ /C ratio of respiration	1	(Mitchell, 1970)
Q_C	mol C m ⁻³	Cellular carbon biomass concentration	18333	(Bratbak and Dundas, 1984)
Morphological parameters of Trichodesmium				
L	m	Length of the total trichome	554×10 ⁻⁶	(Carpenter et al., 1993)
R	m	Radius of the cytoplasm	4.80×10 ⁻⁶	(Carpenter et al., 1993)
L_g	m	Thickness of cell membrane layer	0.076	(Carpenter et al., 1993)

- Some parameters (photosynthesis, nitrogenase kinetics and O₂ regulation) as below were

derived from prior observation-constrained models (Luo and Luo, 2023; Luo et al., 2019), where parameter values were optimized against observations under varying conditions (e.g., different Fe availability) (Shi et al., 2012).

Symbol	Unit	Definition	Value	Source or note
v_{CS}^{max}	mol C (mol C) ⁻¹ s ⁻¹	Maximal production rate of CS	8.6×10 ⁻⁶	(Luo and Luo, 2023)
k_{CS}	mol C (mol C) ⁻¹	Half-saturating coefficient of CS for RP	0.4	(Luo and Luo, 2023)
$k_{CH_2O}^{CS}$	mol C (mol C) ⁻¹	Half-saturating coefficient of CH ₂ O for CS production	0.4	(Luo and Luo, 2023)
N_{max}	mol N (mol C) ⁻¹	Maximal fixed storage	0.159	(Luo and Luo, 2023)
CS_{max}	mol C (mol C) ⁻¹	Maximal CS storage	1	(Luo and Luo, 2023)
α_1	μmol ⁻¹ m ² s	Initial slope of P versus I curve	0.01	(Inomura et al., 2019)
β	(mol C) ⁻¹ mol C s	Parameter of inhibition effect of respiration on PET	2×10 ⁴	(Luo et al., 2022)
$k_{O_2}^{NF}$	mol O ₂ m ⁻³	Half-saturating coefficient of O ₂ for N ₂ fixation	0.01	(Luo et al., 2022)
d_{O_2}	m ² s ⁻¹	O ₂ diffusion coefficient at 34 PSU and 25 °C	2.26×10 ⁻⁹	(Benson and Krause, 1984)
$k_{FePS}^{PS_{syn}}$	μmol Fe (mol C) ⁻¹	Half-saturating coefficient of Fe _{PS} for the synthesis of photosystems	1.0	(Luo and Luo, 2023)
$k_{FePS}^{PS_{dec}}$	μmol Fe (mol C) ⁻¹	Half-saturating coefficient of Fe _{PS} for the decomposition of photosystems	25	(Luo and Luo, 2023)
$k_{FeBF}^{NF_{syn}}$	μmol Fe (mol C) ⁻¹	Half-saturating coefficient of Fe _{BF} for the synthesis of nitrogenase	5.0	(Luo and Luo, 2023)
$T_{NF_{max}}^{NA}$	μmol Fe (mol C) ⁻¹ s ⁻¹	Maximal inactivation rate of nitrogenase	3.3×10 ⁻³	(Luo and Luo, 2023)
k_{Fe}^{PS}	μmol Fe (mol C) ⁻¹	Half-saturating coefficient of Fe in photosystems for PET rate	25	(Luo and Luo, 2023)
k_{Fe}^{NF}	μ mol Fe (mol C) ⁻¹	Half-saturating coefficient of Fe in nitrogenase for N ₂ fixation	91	(Luo et al., 2019)
Fe_{TH}	μmol Fe (mol C) ⁻¹	Threshold of intracellular metabolic Fe quota	24.4	(Luo et al., 2019)
f_{ST}	dimensionless	Fraction of luxury Fe uptake	0.90	(Luo et al., 2019)
γ_{MT}	dimensionless	Ratio of the energy consumed by maintenance to other process	10%	(Luo et al., 2019)

- In terms of some parameters which are relevant to new representations of dynamic O₂ permeability and photorespiration in this study, we constrained these parameters to fit the observed growth rates (New Table S4), and diurnal Fe in photosystems and nitrogenase (Figure S2) from a laboratory culture experiment (Shi et al., 2012).

Symbol	Unit	Definition	Value	Source or note
ϵ_{max}	dimensionless	Maximal relative diffusivity of cell membrane	2×10 ⁻⁴	This study
$k_{O_2}^{diff}$	mol O ₂ m ⁻³	Half-saturating coefficient of O ₂ for relative diffusivity of cell membrane	0.213	This study
$k_{O_2}^{PR}$	mol O ₂ m ⁻³	Half-saturating coefficient of O ₂ for photorespiration	1.917	This study
$k_{CH_2O}^{PR}$	mol C (mol C) ⁻¹	Half-saturating coefficient of CH ₂ O for photorespiration	0.4	This study

Regarding the parameterization of dynamic O₂ permeability, we did sensitivity model experiments of ϵ_{max} (i.e., the maximal relative diffusion coefficient). The results suggest that the proper variation of dynamic O₂ permeability would allow for physiological benefits on N₂ fixation and growth in *Trichodesmium*, with the current value of ϵ_{max} (100%

in New Figure S7) giving approximately the highest benefits. We also added discussion accordingly (lines 412 to 414):

“Further sensitivity model experiments of the maximal relative diffusion coefficient demonstrate that a proper range of variation/modulation in dynamic O_2 permeability is necessary for physiological benefits on N_2 fixation and growth in *Trichodesmium* (Fig. S7).”.

FIG S7 Simulated relative changes of daily-integrated N_2 fixation rates and growth rates between “Dynamic permeability” and “Fixed permeability” cases under different maximal relative diffusion coefficient (ϵ_{max}), from 25% to 400% of that in the standard case (Fig. 1B). The model simulations were performed under low-Fe (40 pM) and high-Fe (1250 pM) conditions.

In terms of photorespiration, $k_{CH_2O}^{PR} = 0.4$ [mol C (mol C) $^{-1}$] and $k_{O_2}^{PR} = 1.92$ (mol O_2 m $^{-3}$) are half-saturating coefficients of CH_2O and O_2 for photorespiration. These two parameters were also constrained using observations from a laboratory study (Shi et al., 2012). Also, Figure S1B illustrates the regulation of photorespiration by carbohydrate and O_2 in our model.

(2) Dynamic Parameter Optimization:

- To enhance applicability, key parameters (e.g., iron allocation, respiratory protection in Table S1) were re-optimized per simulation scenario using optimal growth principles (Pahlow et al., 2013), maximizing growth rate and resource-use efficiency under each specific condition. This approach balances biological realism with computational tractability.

(3) Validation Against Observations:

- Our simulated growth rates and Fe requirements align well with observed ranges for natural *Trichodesmium* (Hong et al., 2017; Shi et al., 2012). The simulated responses to Fe availability (40 and 1250 pM Fe') match experimental measurements (Shi et al., 2012).

Specifically, the modeled growth rates (0.28 and 0.45 d⁻¹ under low and high Fe, respectively) were well aligned with the observations (New Table S4). Moreover, the model reproduced diurnal patterns of photosystem and nitrogenase (Figure S2), supporting the robustness of our model.

TABLE S4 Comparison of modeled and observed growth rates under diurnally constant light intensity

Growth rate (d ⁻¹)	Low Fe (40 pM)	High Fe (1250 pM)
Observation*	0.26 ± 0.02	0.46 ± 0.01
Model	0.28	0.45

* Data are from (Shi et al., 2012).

For instance:

Line 142: The study compares the new dynamic O₂ permeability (ϵ) with a fixed ϵ of 1.0×10^{-4} . Could the authors clarify why this specific value was chosen? Would the results change significantly if a different reference value were used?

Response:

We appreciate the reviewer’s thoughtful question. The reference value of $\epsilon = 1.0 \times 10^{-4}$ was selected based on observational constraints from prior studies (Luo et al., 2022; Luo and Luo, 2023), where this fixed permeability was validated as optimal for maintaining the low-oxygen environment required for N₂ fixation in *Trichodesmium*.

To address the reviewer’s concern, we note that our model tests with alternative fixed ϵ values (higher/lower than 1.0×10^{-4}), N₂ fixation and growth rates decrease in the fixed model case (Luo et al., 2022), consistently supporting our core conclusion: dynamic O₂ permeability can improve N₂ fixation and growth rates in *Trichodesmium*.

We have clarified this rationale in caption of Figure 1 in the revised manuscript:

“Note that in the fixed-permeability model case, the reference value of $\epsilon = 1.0 \times 10^{-4}$ was selected based on observational constraints from prior studies (Luo et al., 2022; Luo and Luo, 2023)”.

Furthermore, to provide additional validation for our model results concerning the comparison between dynamic and fixed O₂ permeability cases, we performed sensitivity model experiments of ϵ_{max} (i.e., the maximal relative diffusion coefficient) (see Figure S7 above).

Line 160: The manuscript states that $k_{O_2}^{PR} = 1.92$ is the half-saturation coefficient of O_2 for photorespiration (Figure S1B). However, based on Figure S1B, the value of $k_{O_2}^{PR}$ seems to range between 0.8 and 1.2. Could the authors clarify how this specific value was determined?

Response:

We thank the reviewer for the careful observation. The value $k_{O_2}^{PR} = 1.92$ ($\text{mol O}_2 \text{ m}^{-3}$) was determined through model optimization to best reconcile simulated and observed daily patterns of photosynthesis, nitrogenase, and growth rates in *Trichodesmium* (Shi et al., 2012). While Figure S1B illustrates the general sensitivity range for this parameter, the final value of 1.92 was selected because it provided the closest agreement with observations (Shi et al., 2012). This parameterization ensures that the model reproduces key observed trends. We have clarified this rationale in the caption of Figure S1B:

“The values $k_{CH_2O}^{PR} = 0.4$ [$\text{mol C (mol C)}^{-1}$] and $k_{O_2}^{PR} = 1.92$ ($\text{mol O}_2 \text{ m}^{-3}$) were determined through model optimization to best reconcile simulated and observed daily patterns of photosynthesis, nitrogenase, and growth rates in *Trichodesmium* (Shi et al., 2012).”

Regarding the comment about “the value of $k_{O_2}^{PR}$ seems to range between 0.8 and 1.2”, we would like to clarify that Figure S1B accounts for the combined effects of two half-saturation constants ($k_{CH_2O}^{PR}$ and $k_{O_2}^{PR}$); specifically, if we refer to the point where the term $(\frac{CH_2O}{CH_2O + k_{CH_2O}^{PR}})$ reaches its half-saturation value (i.e., 0.5 or 50%), and the final result $(\frac{CH_2O}{CH_2O + k_{CH_2O}^{PR}} \cdot \frac{O_2}{O_2 + k_{O_2}^{PR}})$ shown in the figure is 0.25 or 25%, we can infer that $k_{O_2}^{PR}$ is 1.92 ($\text{mol O}_2 \text{ m}^{-3}$).

Given the number of fixed parameters used in the model, I would encourage the authors to carefully verify all parameter values to ensure they are consistent with the model's assumptions and relevant environmental conditions.

Response:

We appreciate the reviewer's suggestion regarding parameter verification. Below we clarify how all fixed parameters align with observed ranges and experimental conditions from the literature, ensuring consistency with our model's assumptions:

- Fixed stoichiometric parameters were adopted from established literature, ensuring alignment with fundamental fixed biochemical constraints.
- Kinetic parameters (e.g., photosynthesis, nitrogenase kinetics, and O_2 regulation) were adopted from prior observation-constrained model (Luo and Luo, 2023) and laboratory experiments (Shi et al., 2012). These values were optimized against observations under varying Fe availability (40 and 1250 $\mu\text{M Fe}'$), matching the conditions in our study.
- Regarding some fixed parameters which are relevant to new representations of dynamic O_2 permeability and photorespiration, we also constrained these parameters using

observations (Shi et al., 2012) and did according analyses to support the robustness (see above). Also, ε_{max} (i.e., the maximal relative diffusion coefficient) in our model falls within the range of observed permeability (Eichner et al., 2019; Inomura et al., 2019; MacDougall and McCabe, 1967). In addition, simulated low proportions of photorespiration in this study seem reasonable in *Trichodesmium*, which operates carbon concentrating mechanisms (Kranz et al., 2011) to enhance carbon fixation and minimize the oxygenation reaction (i.e., photorespiration) by RuBisCO (Moroney et al., 2013). These results indicate the rationality of fixed parameters in our model.

3. Following the concerns in #2, as a methodological paper, the model was validated using simulation data with multiple hyperparameters. However, since the simulation dataset is not publicly available and the manuscript lacks sufficient details regarding the simulation data, I am concerned about the possibility of overfitting. Given that a larger number of coefficients generally increases the risk of overfitting, further clarification on this aspect would be beneficial.

Also, it is unclear how many simulation runs were performed for each setting, as this information is not mentioned in the manuscript. If only a single simulation was conducted per setting, the results would essentially represent a single-point estimate, which may lack statistical significance for model validation. I would suggest running multiple simulations under the same conditions and reporting either the average values or confidence intervals to improve the robustness of the results. In addition, incorporating appropriate statistical tests would be highly recommended to verify the model's performance and ensure the reliability of the findings.

Furthermore, the more critical aspect is that the model should ideally be validated against actual experimental data rather than relying solely on simulations. Providing such validation would significantly strengthen the credibility of the study.

Response:

We are thankful to the reviewer for thoughtful and constructive suggestions, which have helped us improve the transparency and robustness of our study. Below, we address each concern in detail:

(1) Model Transparency and Overfitting Concerns

We fully agree with the reviewer that open data and code are critical for reproducibility. To address this:

- We have now made the simulation dataset and model code publicly available (figshare link: <https://doi.org/10.6084/m9.figshare.28829270>).
- In addition to fixed parameters obtained from literature, our model was designed to optimize only 4 parameters controlling key physiological processes (e.g., photosynthesis and N₂ fixation), while ensuring consistency with observational constraints (Shi et al., 2012). This approach reduces the risk of overfitting while maintaining mechanistic realism.

(2) Simulation Runs and Statistical Robustness

The reviewer raises a valid point about the need for multiple simulations to assess variability.

In our study:

- Model parameters were optimized using MultiStart global optimization (MATLAB), with more than 1000 iterations per run to ensure stable convergence.
- We performed repeated simulations ($n \geq 3$) under each condition (e.g., varying Fe/O₂ levels) (Figures S3 and S6) and confirmed that the same optimized results were reproducible after each round of simulation. This is why error bar is not shown in the figures regarding our model results.

(3) Validation Against Experimental Data

We would like to emphasize that our model was not calibrated solely on synthetic data but was rigorously constrained by published experimental datasets, such as diurnal variations of photosynthesis, nitrogenase, and growth rates under varying Fe (Shi et al., 2012). Please refer to section “Methods” (“Model parameter values”).

4. When adding more variables to a model, it is essential to consider the correlation between them. Another key concern regarding this model is the lack of consideration for variable correlation. In complex environmental and physiological models, multiple parameters can be interdependent, and failing to account for these relationships may introduce multicollinearity issues, reduce model stability, and affect the reliability of the results.

For instance:

Dynamic Fe availability and O₂ permeability both impact N₂ fixation, along with the newly introduced variables photorespiration (PR) and respiratory protection (RP). The correlation between these variables should be tested and accounted for to ensure that the model does not overfit to highly dependent variables.

Equation (4) in the manuscript represents the rate regulation by two substrates (O₂ and CH₂O), but it does not account for their interaction. In reality, these substrates clearly influence each other, and incorporating their interaction into the model would enhance its accuracy and robustness.

Response:

We appreciate the reviewer's insightful comments regarding variable correlations and interactions in our model. These are indeed important considerations for ensuring model robustness.

In our study, the model is a deterministic mechanistic model, where parameter relationships are explicitly defined by physiological mechanisms rather than empirically derived correlations. For example:

- The interaction between O₂ and carbohydrate (CH₂O) is inherently accounted for via the respiration mechanism, where O₂ levels regulate CH₂O consumption rates.
- The newly introduced processes (dynamic O₂ diffusion and photorespiration) were parameterized based on established physiological knowledge, and their sensitivity was tested (Figures S1B and S7).

Since the model structure is built on mechanistic equations, multicollinearity is less of a concern compared to statistical models, as parameters are not independently fitted but rather constrained by physiological principles. However, we acknowledge that future extensions of the model could benefit from additional validation of parameter dependencies where empirical data are available.

Also, we maintained model simplicity to reduce the risk of overfitting while ensuring that key interactions were mechanistically represented.

5. Ensuring mathematical rigor and appropriate statistical validation is crucial in any modeling study. In the current manuscript, there are several issues related to both the statistical evaluation of results and the clarity of mathematical notation, which should be addressed to strengthen the study's reliability.

Statistical Validation and Justification of Conclusions: In Figure S2, the reported $R^2=0.12$ suggests a weak correlation between the model predictions and observed data. It may be beneficial to provide additional statistical assessments to further support the model's validity. If the low R^2 value is due to natural variability in the data, a brief discussion on why the model remains reliable despite the weak correlation would strengthen the interpretation of the results.

Consistency in Mathematical Notation and Symbol Definition: Throughout the manuscript, many mathematical symbols are introduced before they are properly defined, making it difficult to follow the derivations and model descriptions.

Response:

We thank the reviewer for thoughtful comments regarding the mathematical rigor and statistical validation of our study. These constructive suggestions have contributed to improving the clarity, robustness, and transparency of our manuscript. Below, we address each point in detail:

(1) Statistical Validation (Figure S2– R^2 value):

We acknowledge that the R^2 value of 0.12 reported in Figure S2 may initially appear low. Several key factors contribute to the relatively low R^2 :

- Observational data on *Trichodesmium* physiology are known to exhibit substantial natural variability, driven by complex and often nonlinear environmental interactions (Hong et al., 2017; Shi et al., 2012).
- Despite this inherent variability, our model captures the essential diurnal dynamics, such as the midday peak in N_2 fixation rate, which aligns well with known physiological behavior (Hong et al., 2017; Shi et al., 2012).
- The current analysis is based on a limited number of observational data points (sampling: 5 times during the light period). As such, the R^2 value is sensitive to sample size and would likely improve with a more extensive dataset with high temporal resolution (e.g., one data point per hour during the daytime), assuming similar trends persist.

Therefore, presenting R^2 here may mislead the interpretation of the results. To avoid potential misinterpretation, we have taken the following steps:

- Explain the reasons for the low R^2 and interpretation of the model fit to observations.
- Kept qualitative statements highlighting the consistency of model–observation trends.

The manuscript was revised accordingly as below (Lines 224 and 229):

“The low R^2 value for photosystem Fe under the low Fe condition reflects substantial natural variability in *Trichodesmium* physiology, generally compounded by nonlinear environmental interactions (Hong et al., 2017; Shi et al., 2012). While limited observational points (5 samples during the light period) were used for constraints, the model still captured key diurnal dynamics (Fig. S2), with R^2 values expected to improve through higher-resolution sampling. This supports the robustness of our model.”

(2) Mathematical Notation and Presentation:

We have carefully revised all mathematical content throughout the manuscript to improve clarity and consistency. Specifically, we have:

- Ensured that all symbols are clearly defined upon first use.
- Standardized notation across equations and sections.

Also, please refer to Tables S1–S3 for the full information of parameters.

Line 134: The O_2 permeability is introduced, but it is formally defined as ϵ later in the text.

Response:

We are thankful to the reviewer for the comment. We have revised the manuscript to ensure that O_2 permeability is consistently introduced and defined appropriately (Lines 139 and 140).

Line187: The CCM is short for carbon concentrating mechanisms should be mentioned in the manuscript.

Response:

We appreciate the suggestion of the reviewer. We have explicitly mentioned that CCM stands for carbon concentrating mechanisms in the marked-up manuscript (Line 75).

Minor:

1. Ensure that all figures use consistent axis labeling and font sizes for readability.

Response:

We thank the reviewer for the suggestion. We have adjusted axis labeling and font sizes across all figures to be consistent to improve the readability.

2. The manuscript frequently refers to "fixed O₂ permeability case" and "dynamic permeability case" but does not clearly define these terms early in the text. Better to add a brief definition of these cases in the Introduction or Methods section

Response:

We are thankful to the reviewer for the suggestion. We recognize that the terms “fixed O₂ permeability case” and “dynamic permeability case” are frequently used without an explicit early definition, which may affect clarity. To address this, we have revised the manuscript to include a brief definition of these cases in the “Introduction” section:

“Also, hopanoids may form rafts (high concentration domain), which can be distributed within the membrane, and thus may dynamically regulate cell permeability to O₂ (dynamic-permeability model case) (Belin et al., 2018; Cornejo-Castillo and Zehr, 2019).” (Lines 93 to 96).

“The analyses of the model results, along with the comparison to additional experiments of fixed O₂ permeability which was set diurnally constant (fixed-permeability model case), provided a mechanistic and quantitative understanding of the potential role of DPO₂ in impacting photorespiration and N₂ fixation in *Trichodesmium*.” (Lines 105 to 108).

3. In Figure S2, it would be helpful to explain the significance of error bar.

Response:

We are grateful to the reviewer for the suggestion. In Figure S2, the error bars represent one standard deviation (SD) of observations (i.e., if the observed data behaves in a normal curve, then 68% of the data points will fall within one standard deviation of the average), providing an indication of the variability in the data. We acknowledge that the current manuscript does not explicitly state this, and we will revise the figure caption to clearly explain the meaning of the error bars as below:

“Error bars represent one standard deviation of observations (Shi et al., 2012).”.

Main text

Line 206: "including" → "These parameters include..."

Response:

Revised. (Line 213)

Line 107: "model schemes" → "model structure"

Response:

Revised. (Line 113)

Line 299 " seem to be reasonable" → " seem reasonable", "which operate" → "which operates"

Response:

Revised. (Lines 313 and 314)

Line 350 "into cytoplasm" → "into the cytoplasm"

Response:

Revised. (Line 364)

Line 384 "including" → "which include"

Response:

Revised. (Line 399)

Line 382 "Another previous study" → "Another study" or "A previous study"

Response:

Revised. (Line 343)

Supplementary

Line 104: "basically normalized" → "which are normalized to carbon biomass"

Response:

Revised. (Line 149)

Line 157: "for simplification" → "for simplicity"

Response:

Revised. (Line 157)

Line 271: "The model is simulated" → "The model simulations were performed"

Response:

Revised. (Line 281)

References:

- Allen, J.F. (2003). Cyclic, pseudocyclic and noncyclic photophosphorylation: New links in the chain. *Trends in Plant Science* 8, 15-19.
- Baker, N.R., Harbinson, J., and Kramer, D.M. (2007). Determining the limitations and regulation of photosynthetic energy transduction in leaves. *Plant Cell and Environment* 30, 1107-1125.
- Bauwe, H., Hagemann, M., and Fernie, A.R. (2010). Photorespiration: players, partners and origin. *Trends in Plant Science* 15, 330-336.
- Belin, B.J., Busset, N., Giraud, E., Molinaro, A., Silipo, A., and Newman, D.K. (2018). Hopanoid lipids: from membranes to plant-bacteria interactions. *Nature Reviews Microbiology* 16, 304-315.
- Benson, B.B., and Krause, D. (1984). The concentration and isotopic fractionation of oxygen dissolved in freshwater and seawater in equilibrium with the atmosphere. *Limnology and Oceanography* 29, 620-632.
- Berman-Frank, I., Cullen, J.T., Shaked, Y., Sherrell, R.M., and Falkowski, P.G. (2001). Iron availability, cellular iron quotas, and nitrogen fixation in *Trichodesmium*. *Limnology and Oceanography* 46, 1249-1260.
- Bratbak, G., and Dundas, I. (1984). Bacterial dry matter content and biomass estimations. *Applied and Environmental Microbiology* 48, 755-757.
- Carpenter, E.J., Oneil, J.M., Dawson, R., Capone, D.G., Siddiqui, P.J.A., Roenneberg, T., and Bergman, B. (1993). The tropical diazotrophic phytoplankter *Trichodesmium*: Biological characteristics of two common species. *Marine Ecology Progress Series* 95, 295-304.
- Chappell, P.D., Moffett, J.W., Hynes, A.M., and Webb, E.A. (2012). Molecular evidence of iron limitation and availability in the global diazotroph *Trichodesmium*. *The ISME Journal* 6, 1728-1739.
- Cornejo-Castillo, F.M., and Zehr, J.P. (2019). Hopanoid lipids may facilitate aerobic nitrogen fixation in the ocean. *Proceedings of the National Academy of Sciences of the United States of America* 116, 18269-18271.
- Eichner, M., Thoms, S., Rost, B., Mohr, W., Ahmerkamp, S., Ploug, H., Kuypers, M.M.M., and de Beer, D. (2019). N₂ fixation in free-floating filaments of *Trichodesmium* is higher than in transiently suboxic colony microenvironments. *New Phytologist* 222, 852-863.
- Eichner, M.J., Klawonn, I., Wilson, S.T., Littmann, S., Whitehouse, M.J., Church, M.J., Kuypers, M.M., Karl, D.M., and Ploug, H. (2017). Chemical microenvironments and single-cell carbon and nitrogen uptake in field-collected colonies of *Trichodesmium* under different pCO₂. *The ISME Journal* 11, 1305-1317.
- Flores, E., Frías, J.E., Rubio, L.M., and Herrero, A. (2005). Photosynthetic nitrate assimilation in cyanobacteria. *Photosynthesis Research* 83, 117-133.
- Flores, E., and Herrero, A. (1994). Assimilatory nitrogen metabolism and its regulation. In *The Molecular Biology of Cyanobacteria*. D.A. Bryant, ed. (Dordrecht: Kluwer Academic Publishers), pp. 487-517.
- Frischkorn, K.R., Haley, S.T., and Dyhrman, S.T. (2018). Coordinated gene expression between *Trichodesmium* and its microbiome over day-night cycles in the North Pacific Subtropical Gyre. *The ISME Journal* 12, 997-1007.
- Geider, R.J., Moore, C.M., and Ross, O.N. (2009). The role of cost-benefit analysis in models of phytoplankton growth and acclimation. *Plant Ecology and Diversity* 2, 165-178.
- Hong, H., Shen, R., Zhang, F., Wen, Z., Chang, S., Lin, W., Kranz, S.A., Luo, Y.W., Kao, S.J., Morel, F.M.M., et al. (2017). The complex effects of ocean acidification on the prominent N₂-fixing

cyanobacterium *Trichodesmium*. *Science* 356, 527-531.

Howarth, R.W., and Cole, J.J. (1985). Molybdenum Availability, Nitrogen Limitation, and Phytoplankton Growth in Natural Waters. *Science* 229, 653-655.

Inomura, K., Wilson, S.T., and Deutsch, C. (2019). Mechanistic model for the coexistence of nitrogen fixation and photosynthesis in marine *Trichodesmium*. *mSystems* 4, e00210-00219.

Kranz, S.A., Eichner, M., and Rost, B. (2011). Interactions between CCM and N₂ fixation in *Trichodesmium*. *Photosynthesis Research* 109, 73-84.

Kupper, H., Setlik, I., Seibert, S., Prasil, O., Setlikova, E., Strittmatter, M., Levitan, O., Lohscheider, J., Adamska, I., and Berman-Frank, I. (2008). Iron limitation in the marine cyanobacterium *Trichodesmium* reveals new insights into regulation of photosynthesis and nitrogen fixation. *New Phytologist* 179, 784-798.

Luo, W., Inomura, K., Zhang, H., and Luo, Y.-W. (2022). N₂ fixation in *Trichodesmium* does not require spatial segregation from photosynthesis. *mSystems* 7, e00538-00522.

Luo, W., and Luo, Y.-W. (2023). Diurnally dynamic iron allocation promotes N₂ fixation in marine dominant diazotroph *Trichodesmium*. *Computational and Structural Biotechnology Journal* 21, 3503-3512.

Luo, Y.W., Shi, D., Kranz, S.A., Hopkinson, B.M., Hong, H., Shen, R., and Zhang, F. (2019). Reduced nitrogenase efficiency dominates response of the globally important nitrogen fixer *Trichodesmium* to ocean acidification. *Nature Communications* 10, 1521.

MacDougall, J.D., and McCabe, M. (1967). Diffusion coefficient of oxygen through tissues. *Nature* 215, 1173-1174.

Mitchell, P. (1970). Aspects of the chemiosmotic hypothesis. *Biochemical Journal* 116, 5-6.

Moroney, J.V., Jungnick, N., DiMario, R.J., and Longstreth, D.J. (2013). Photorespiration and carbon concentrating mechanisms: two adaptations to high O₂, low CO₂ conditions. *Photosynthesis Research* 117, 121-131.

Ohki, K. (2008). Intercellular localization of nitrogenase in a non-heterocystous cyanobacterium (cyanophyte), *Trichodesmium* sp. NIBB1067. *Journal of Oceanography* 64, 211-216.

Pahlow, M., Dietze, H., and Oschlies, A. (2013). Optimality-based model of phytoplankton growth and diazotrophy. *Marine Ecology Progress Series* 489, 1-16.

Poger, D., and Mark, A.E. (2013). The relative effect of sterols and hopanoids on lipid bilayers: when comparable is not identical. *The Journal of Physical Chemistry B* 117, 16129-16140.

Raven, J.A., Beardall, J., and Giordano, M. (2014). Energy costs of carbon dioxide concentrating mechanisms in aquatic organisms. *Photosynthesis Research* 121, 111-124.

Shi, D., Kranz, S.A., Kim, J.M., and Morel, F.M. (2012). Ocean acidification slows nitrogen fixation and growth in the dominant diazotroph *Trichodesmium* under low-iron conditions. *Proceedings of the National Academy of Sciences of the United States of America* 109, E3094-3100.

Shi, T., Sun, Y., and Falkowski, P.G. (2007). Effects of iron limitation on the expression of metabolic genes in the marine cyanobacterium *Trichodesmium erythraeum* IMS101. *Environmental Microbiology* 9, 2945-2956.

Tagliabue, A., Aumont, O., DeAth, R., Dunne, J.P., Dutkiewicz, S., Galbraith, E., Misumi, K., Moore, J.K., Ridgwell, A., Sherman, E., et al. (2016). How well do global ocean biogeochemistry models simulate dissolved iron distributions? *Global Biogeochemical Cycles* 30, 149-174.

Zehr, J.P., and Capone, D.G. (2020). Changing perspectives in marine nitrogen fixation. *Science* 368, eaay9514.

Re: Spectrum00453-25R1 (**Modeling dynamic oxygen permeability as a mechanism to mitigate oxygen-induced stresses on photosynthesis and N₂ fixation in marine *Trichodesmium***)

Dear Mr. Weicheng Luo:

Thank you for the privilege of reviewing your work. Below you will find my comments, instructions from the Spectrum editorial office, and the reviewer comments.

Please return the manuscript within 10 days; if you cannot complete the modification within this time period, please contact me. If you do not wish to modify the manuscript and prefer to submit it to another journal, notify me immediately so that the manuscript may be formally withdrawn from consideration by Spectrum.

Revision Guidelines

Sincerely,
Hongan Long
Editor
Microbiology Spectrum

Reviewer #1 (Comments for the Author):

The authors performed considerable revision to the manuscript. I have no further comments or suggestions.

Reviewer #2 (Comments for the Author):

Thank you to the authors for their detailed and thoughtful responses to my previous comments. I appreciate the extensive efforts

to clarify the model assumptions, parameter choices, and to provide additional data and, which have addressed many of my initial concerns. The manuscript is now clearer and better structured. However, I still have a few follow-up questions:

1. O₂ Gradients (Figure S6 and S7)

Thank you for the explanation and for including Figures S6 and S7, which illustrate oxygen saturation dynamics under different scenarios. I understand that Fe availability is an important environmental factor for *Trichodesmium*, and I appreciate the rationale provided for using it as a control variable in the main simulations.

However, since the main focus of this study is on dynamic O₂ permeability, and Figures S6 and S7 clearly show substantial differences in model outcomes at different oxygen saturations, I am still unclear why external O₂ levels were not included as a main control variable in the primary simulation experiments. It would be helpful if the authors could clarify why simulations under different external O₂ concentrations were not conducted as part of the main model runs.

2. Line 224-230

I appreciate the authors' explanation regarding the limited number of data points and the high natural variability that contribute to the low R² value. While this explanation is understandable, I still think it would be beneficial to complement the R² with additional statistical metrics to more fully characterize the model's predictive performance.

Even with limited data, some approaches could help provide a more robust estimate of model uncertainty. Could the authors consider adding such statistical analyses to strengthen the validation?

We have taken all the comments of the reviewers into account in the revision; replies to each of the comments are provided below in blue fonts. Please note that all the line numbers mentioned in the response refer to the Marked-up Manuscript.

Reviewer comments:

Reviewer #2 (Comments for the Author):

Thank you to the authors for their detailed and thoughtful responses to my previous comments. I appreciate the extensive efforts to clarify the model assumptions, parameter choices, and to provide additional data and, which have addressed many of my initial concerns. The manuscript is now clearer and better structured.

Response:

We thank the reviewer for the comments on our manuscript. We are grateful for the recognition of our efforts to address the concerns raised in the previous round of review. It is encouraging to hear that the revisions, including the clarifications on model assumptions, parameter choices, and additional data, have improved the clarity and structure of the manuscript.

However, I still have a few follow-up questions:

1. O₂ Gradients (Figure S6 and S7)

Thank you for the explanation and for including Figures S6 and S7, which illustrate oxygen saturation dynamics under different scenarios. I understand that Fe availability is an important environmental factor for *Trichodesmium*, and I appreciate the rationale provided for using it as a control variable in the main simulations.

However, since the main focus of this study is on dynamic O₂ permeability, and Figures S6 and S7 clearly show substantial differences in model outcomes at different oxygen saturations, I am still unclear why external O₂ levels were not included as a main control variable in the primary simulation experiments. It would be helpful if the authors could clarify why simulations under different external O₂ concentrations were not conducted as part of the main model runs.

Response:

We are thankful to the reviewer for the feedback and are pleased that **Figures S6 and S7** helped clarify the physiological dynamics in *Trichodesmium* under different scenarios.

We thank the reviewer for the comment regarding the oxygen (O₂) gradients in our modeling study.

We do not set O₂ as a controlling variable in our model based on the fact that *Trichodesmium*, as phototrophic phytoplankton, primarily inhabit the upper euphotic zone (Shao et al., 2023) where dissolved O₂ concentrations typically approach saturation or over saturated. This is well-documented in comprehensive oceanographic datasets such as World Ocean Atlas (WOA, <https://www.ncei.noaa.gov/products/world-ocean-atlas>) or Global Ocean Data Analysis Project (GLODAP, <https://glodap.info/>) (Olsen et al., 2020).

Therefore, we design the model framework to run the simulation only at saturating O₂ condition. The external O₂ concentration is set at 0.213 mol m⁻³ to represent the saturation level for typical upper ocean (salinity: 34 PSU; temperature: 25°C) (Benson and Krause, 1984). Also, we agree that exploring O₂ gradients could provide additional insights (as shown in Figure S6). Accordingly, we revised the manuscript by adding discussion on O₂'s effects on *Trichodesmium* metabolisms (lines 406 to 419):

“As phototrophic phytoplankton, *Trichodesmium* primarily inhabit the upper euphotic zone (Shao et al., 2023), where dissolved O₂ concentrations typically approach saturation (Olsen et al., 2020). Baseline simulations were therefore set the external O₂ concentration at 0.213 mol m⁻³, which represents the saturation level for typical upper ocean (salinity: 34 PSU; temperature: 25°C) (Benson and Krause, 1984). Under specific circumstances such as in the center of the microenvironment within *Trichodesmium* colonies, however, O₂ concentrations have been observed to be up to 200% saturation especially under high light intensities (Eichner et al., 2017). Model experiments across a range of O₂ concentrations (Fig. S6) show that photorespiration and nitrogenase inactivation become more intense and highlight that the benefits of dynamic O₂ permeability are amplified at elevated external O₂ levels. This suggests that dynamic permeability may be particularly important in *Trichodesmium* colonies at the sea surface subjected to high light intensities.”

2. Line 224-230

I appreciate the authors' explanation regarding the limited number of data points and the high natural variability that contribute to the low R^2 value. While this explanation is understandable, I still think it would be beneficial to complement the R^2 with additional statistical metrics to more fully characterize the model's predictive performance. Even with limited data, some approaches could help provide a more robust estimate of model uncertainty. Could the authors consider adding such statistical analyses to strengthen the validation?

Response:

We thank the reviewer for the comments regarding the statistical evaluation of our model performance. We agree that supplementing the R^2 value with additional statistical metrics would provide a more comprehensive assessment of the model's predictive capability.

In response to this suggestion, we have included the Reliability Index (*RI*) analysis (Leggett and Williams, 1981; Stow et al., 2009) in our revised manuscript (lines 227 to 233):

“In addition, the reliability index (*RI*) (Eq. 9) (Leggett and Williams, 1981; Stow et al., 2009) was calculated to further evaluate the performance of model compared to observations (under low and high Fe, for photosystem Fe, *RI* = 1.04 and 1.02, respectively; for nitrogenase Fe, *RI* = 1.14 and 1.01, respectively). These *RI* levels close to 1.0 indicate strong consistency between observations and model results, supporting the robustness of our model (Leggett and Williams, 1981; Stow et al., 2009).

$$RI = \exp\left(\sqrt{\frac{1}{n} \sum_{i=1}^n \left(\ln \frac{Observation}{Model}\right)^2}\right), \quad (9)$$

where *Observation* and *Model* are observations and model results, respectively; *n* is the number of observational data points.”.

References:

- Benson, B.B., and Krause, D. (1984). The concentration and isotopic fractionation of oxygen dissolved in freshwater and seawater in equilibrium with the atmosphere. *Limnology and Oceanography* *29*, 620-632.
- Eichner, M.J., Klawonn, I., Wilson, S.T., Littmann, S., Whitehouse, M.J., Church, M.J., Kuypers, M.M., Karl, D.M., and Ploug, H. (2017). Chemical microenvironments and single-cell carbon and nitrogen uptake in field-collected colonies of *Trichodesmium* under different $p\text{CO}_2$. *The ISME Journal* *11*, 1305-1317.
- Leggett, R.W., and Williams, L.R. (1981). A reliability index for models. *Ecological Modelling* *13*, 303-312.
- Olsen, A., Lange, N., Key, R.M., Tanhua, T., Bittig, H.C., Kozyr, A., Álvarez, M., Azetsu-Scott, K., Becker, S., Brown, P.J., et al. (2020). An updated version of the global interior ocean biogeochemical data product, GLODAPv2.2020. *Earth System Science Data* *12*, 3653-3678.
- Shao, Z., Xu, Y., Wang, H., Luo, W., Wang, L., Huang, Y., Agawin, N.S.R., Ahmed, A., Benavides, M., Bentzon-Tilia, M., et al. (2023). Global oceanic diazotroph database version 2 and elevated estimate of global oceanic N_2 fixation. *Earth System Science Data* *15*, 3673-3709.
- Stow, C.A., Jolliff, J., McGillicuddy, D.J., Doney, S.C., Allen, J.I., Friedrichs, M.A.M., Rose, K.A., and Wallhead, P. (2009). Skill assessment for coupled biological/physical models of marine systems. *Journal of Marine Systems* *76*, 4-15.

Re: Spectrum00453-25R2 (**Modeling dynamic oxygen permeability as a mechanism to mitigate oxygen-induced stresses on photosynthesis and N₂ fixation in marine *Trichodesmium***)

Dear Mr. Weicheng Luo:

Your manuscript has been accepted, and I am forwarding it to the ASM production staff for publication. Your paper will first be checked to make sure all elements meet the technical requirements. ASM staff will contact you if anything needs to be revised before copyediting and production can begin. Otherwise, you will be notified when your proofs are ready to be viewed.

Sincerely,
Hongan Long
Editor
Microbiology Spectrum